# INTROSPECTIVE LEARNING : A TWO-STAGE APPROACH FOR INFERENCE IN NEURAL NETWORKS

## ABSTRACT

In this paper, we advocate for two stages in a neural network's decision making process. The first is the existing feed-forward inference framework where patterns in given data are sensed and associated with previously learned patterns. The second stage is a slower reflection stage where we ask the network to reflect on its feed-forward decision by considering and evaluating all available choices. Together, we term the two stages as introspective learning. We use gradients of trained neural networks as a measurement of this reflection. We perceptually visualize the *post-hoc* explanations from both stages to provide a visual grounding to introspection. For the application of recognition, we show that an introspective network is $4\%$ more robust and $42\%$ less prone to calibration errors when generalizing to noisy data. We also illustrate the value of introspective networks in downstream tasks that require generalizability and calibration including active learning and out-of-distribution detection. Finally, we ground the proposed machine introspection to human introspection in the application of image quality assessment.

## 1 INTRODUCTION

Introspection is the act of looking into one's own mind (Boring, 1953). Classical introspection has its roots in philosophy. Locke (1847), the founder of empiricism, held that all human ideas come from experience. This experience is a result of both sensation and reflection. By sensation, one receives passive information using the sensory systems of sight, sound, and touch. Reflection is the objective observation of our own mental operations. Consider the task of differentiating a spoonbill from a flamingo and a crane. This task requires prior knowledge of some differentiating features between the birds. These features include the color and shape of the body, and beak of all birds. We first associate these features with our existing knowledge of birds and make a coarse decision that the given bird is a spoonbill. This is the sensing stage. Reflection involves questioning the coarse decision and asking why the bird cannot be a flamingo or crane. If the answers are satisfactory, then an introspective decision that the bird is indeed a spoonbill is made. The observation of this reflection is introspection.

In this paper, we adopt this differentiation between sensing and reflection to advocate for two-stage neural network architectures for perception-based applications. We first ground introspection based on existing neural networks. The above-mentioned task of differentiating a spoonbill from a flamingo and crane is provided in Fig. 1 for neural networks A network $f(\cdot)$, is trained on a distribution $\mathcal{X}$ to classify data into $N$ classes. The network learns notions about data samples when classifying them. These notions are stored as network weights $W$. Let $y_{feat}$ be the logits projected before the final fully connected layer. We denote the final fully connected layer as $f_L$, where $L$ is the layer number. $f_{L-1}$ is then the layer before the final fully connected layer. Using the weight parameters $W_L$, the output of the network $\hat{y}$ is given by,

$$y_{feat} = f_{L-1}(x), \forall y_{feat} \in \Re^{N \times 1},$$
$$\hat{y} = \arg\max(W_L^T y_{feat}), \forall W_L \in \Re^{d_{L-1} \times N}, f_{L-1}(x) \in \Re^{d_{L-1} \times 1}. \tag{1}$$

Hence, $\hat{y}$ is the class in which the sensed features maximally correlate with the stored notion. This is the feed-forward prediction in Fig. 1. Existing recognition architectures including VGG (Simonyan & Zisserman, 2015), ResNet (He et al., 2016), and DenseNet (Huang et al., 2017) among others all sense and predict using Eq.1. In Fig. 1, we depict our proposed introspective learning framework. An additional reflection stage extracts the introspective features as *Not Detect* features from $x$. Let

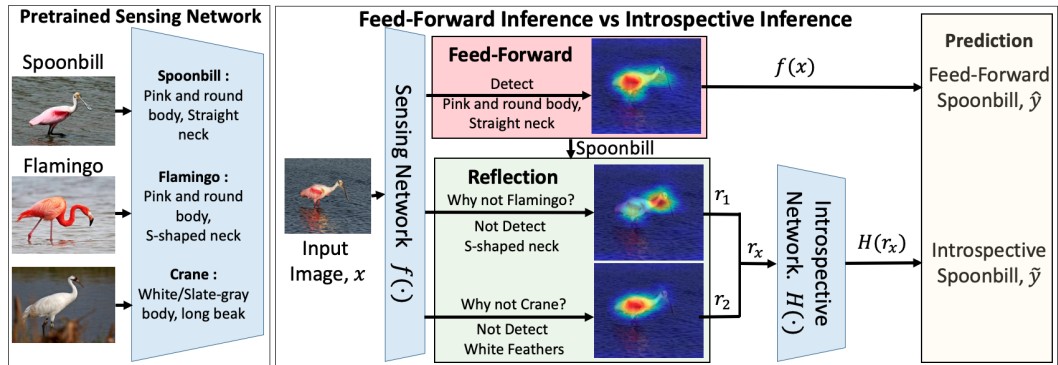

Figure 1: Toy example of feed-forward and introspection process. The visual *post-hoc* explanations in the sensing is from Grad-CAM (Selvaraju et al., 2017) while the explanations in the reflection stage are our own. The written text is for illustrative purpose only.

$r_1$ and $r_2$ be the two introspective features. In this case, $r_1$ is the absence of the S-shaped neck in the spoonbill. And $r_2$ is the lack of white feathers in the given input image. We use a *post-hoc* visual explanation to depict these features[1]. Note that there can be $N$ such features for a sensing network $f(\cdot)$ trained to differentiate between $N$ classes. These features are then combined to obtain the final introspective feature $r_x$. $r_x$ is characteristic of the input image $x$ and is passed through an introspective network, $\mathcal{H}(\cdot)$, to obtain the introspective prediction $\tilde{y}$. We term the combination of both $f(\cdot)$ and $\mathcal{H}(\cdot)$ as introspective learning.

Note that the proposed reflection stage does not require a predefined knowledge base of introspective features. Rather, $r_i, i \in [1, N]$ in the reflection stage, is extracted using the feed-forward prediction $\hat{y}$ and the sensing network parameters. Hence, $r_i$ are *Not Detect* features based on $f(\cdot)$'s notion of classes. $\mathcal{H}(r_x)$ predicts $\tilde{y}$ explicitly based on the differences between $f(\cdot)$'s notion of classes. Not only should the network sense the feed-forward patterns, it must also satisfy $\mathcal{H}(\cdot)$'s $N$ notions of differences. In this paper, we show that the inference process is more generalizable due to these $N$ additional inferential constraints. Specifically, the introspective network is more robust to noise and is less prone to calibration errors. The challenge is to implicitly extract features that answer introspective questions without explicitly training on said questions as is the norm in Visual Question Answering applications (Antol et al., 2015). We show that gradients w.r.t network parameters store notions about the difference between classes and can be used as introspective features. We first describe the methodology of introspective feature extraction in Section 2. We then analyze $\mathcal{H}(\cdot)$ in Section. 3. We show that $\mathcal{H}(\cdot)$ as a simple multi-layer perceptron that introspects on $\hat{y}$ is more generalizable for the application of recognition in Section 5. We then illustrate the benefits of our two-stage architecture in other downstream tasks including out-of-distribution detection, active learning and image quality assessment in Section 6.

## 2 INTROSPECTIVE FEATURES

In this section, we describe introspective features and implicitly extract them using the sensing network. We then analyze their extraction procedure and provide a methodology to accelerate it.

**Definition 2.1** (Introspection). *Given a network $f(\cdot)$, a datum $x$, and the network's prediction $f(x) = \hat{y}$, introspection in $f(\cdot)$ is the measurement of change induced in the network parameters when a label $y_I$ is introduced as the label for $x$. This measurement is the gradient induced by a loss function $J(y_I, \hat{y})$, w.r.t. the network parameters.*

This definition for introspection is in accordance with the sensing and reflection stages in Fig. 1. The network's prediction $\hat{y}$ is the output of the sensing stage and the change induced by an introspective label, $y_I$, is the network reflecting on its decision $\hat{y}$ as opposed to $y_I$. Combination of the two is introspection. Note that introspection can occur when $\hat{y}$ is contrasted against any trained label $y_I, I \in [1, N]$. For instance, in Fig. 1, the network is asked to reflect on its decision of spoonbill by considering other $y_I$ that $x$ can take - flamingo and crane.

---

[1]*post-hoc* explanations are justifications made by a neural network after a decision has been made. They require human interpretation. Further details are provided in Appendix A.

Reflection is the empirical risk that the network has predicted $x$ as $\hat{y}$ instead of $y_I$. Given the network parameters, this risk is measured through some loss function $J(y_I, \hat{y})$. $y_I$ is a one-hot vector with a one at the $I^{th}$ location. The change that is induced in the network is given by the gradient of $J(y_I, \hat{y})$ w.r.t. the network parameters. In this paper, we introspect based on reflecting on all possible classes. For an $N$-class classifier, there are $N$ possible introspective classes and hence $N$ possible gradients each given by, $r_I = \nabla_W J(y_I, \hat{y}), I \in [1, N]$. Here, $r_I$ are the introspective features. Since we introspect based on classes, we measure the change in network weights in the final fully connected layer. Hence the introspective features are given by,

$$r_I = \nabla_{W_L} J(y_I, \hat{y}), I \in [1, N], r_I \in \Re^{d_{L-1} \times N} \tag{2}$$

where $W_L$ are the network weights for the final fully connected layer. Note that the final fully connected layer from Eq. 1 has a dimensionality of $\Re^{d_{L-1} \times N}$. For every $x$, Eq. 2 is applied $N$ times to obtain $N$ separate $r_I$. We first analyze these features before accelerating their extraction.

## 2.1 INTROSPECTIVE FEATURE ANALYSIS

Consider the extraction process in Eq. 2. Each $r_I$ is a $d_{L-1} \times N$ matrix. Expressing gradients in $r_I$ separately w.r.t. the different filters in $W_L$, we have a row-wise concatenated set of gradients given by,

$$r_I = [\nabla_{W_{L,1}} J(y_I, \hat{y}); \nabla_{W_{L,2}} J(y_I, \hat{y}); \nabla_{W_{L,3}} J(y_I, \hat{y}) \dots \nabla_{W_{L,N}} J(y_I, \hat{y})] \tag{3}$$

where each $W_{L,j} \in \Re^{d_{L-1} \times 1}$ and $r_I \in \Re^{d_{L-1} \times N^2}$. For all data $x \in \mathcal{X}$ the following lemma holds:

**Lemma 1.** *Given a unique ordered pair $(x, \hat{y})$ and a trained network $f(\cdot)$, the gradients for a loss function $J(y_I, \hat{y})$ w.r.t. classes are pairwise orthogonal under the second-order Taylor series approximation, each class paired with the predicted class.*

*Proof.* Provided in Appendix B.1. □

Lemma 1 states that backpropagating class $y_I$ does not provide any information to $W_{L,j}, j \neq I$ and hence there is no need to use $\nabla_{W_{L,j}} J(y_j, \hat{y}), j \neq i$ as features when considering $y_I$. In Appendix B.1, we provide the complete proof when $J(y_i, \hat{y})$ is the cross entropy loss. $\nabla_W J(y_I, \hat{y})$ for an introspective class reduces to,

$$\nabla_W J(y_I, \hat{y}) = -\nabla_W y_I + \nabla_W \log\left(\frac{y_{\hat{y}}^2}{2}\right). \tag{4}$$

where $y_{\hat{y}}$ is the logit associated with the predicted class. In Fig. 5, we use a network trained on MNIST (LeCun et al., 1998) dataset to simulate a well-trained network and we visualize the gradients from the final fully connected layer to demonstrate Eq. 4.

Eq. 4 motivates the generalizable nature of our introspective features. Consider some noise added to $x$. To change the prediction $\hat{y}$, the noise must sufficiently decrease $y_{\hat{y}}$ from Eq. 4 and increase the closest logit value, $y_I$, to change the prediction. However, by constraining our final prediction $\tilde{y}$ from Fig. 1 on $N$ such Eq. 4, the noise needs to change the orthogonal relationship between $N$ pairwise logits. This motivates a function $\mathcal{H}(\cdot)$ that is conditioned on $N$ such pairwise logits. In Section. 5, we empirically show the robustness of our feature set.

## 2.2 INTROSPECTIVE FEATURE EXTRACTION

From Lemma 1, the introspective feature is only dependent on the predicted class $\hat{y}$ and the introspective class $y_I$ making their span orthogonal to all other gradients. Hence

$$r_I = \nabla_{W_{L,I}} J(y_I, \hat{y}), I \in [1, N], r_I \in \Re^{d_{L-1} \times 1} \tag{5}$$

Compare Eq. 5 against the introspective feature from Eq. 2. Assuming that forward and backward passes through the final layer $f_L(\cdot)$ are each of $\mathcal{O}(1)$ time complexity, the feed-forward prediction for a given $x$ is $\mathcal{O}(1)$ time complex. Given that $f(\cdot)$ is trained to classify between $N$ classes, extracting

$N$ introspective features require $N$ backpropagations and hence is $\mathcal{O}(N)$ complex. Each $r_I$ in Eq. 2 has a dimensionality $d_{L-1} \times N$. Hence for $N$ features, the space complexity is $\mathcal{O}(N^2 \times d_{L-1})$. From Lemma 1, the introspective feature is only dependent on the predicted class $\hat{y}$ and the introspective class $y_I$ making their span orthogonal to all other gradients. For $N$ introspective features in Eq. 5, the space complexity of $r_I$ reduces from $\mathcal{O}(d_{L-1} \times N^2)$ to $\mathcal{O}(d_{L-1} \times N)$. Note that the bottleneck in time complexity for $N$ gradient extractions are the serial $N$ backpropagations in Eq. 3. Building on Lemma 1, we present the following theorem.

**Theorem 1.** *Given a unique ordered pair $(x, \hat{y})$ and a trained network $f(\cdot)$, the gradients for a loss function $J(y_I, f(x)), I \in [1, N]$ w.r.t. classes when $y_I$ are $N$ orthogonal one-hot vectors is equivalent to when $y_I$ is a vector of all ones, under the second-order Taylor series approximation.*

*Proof.* Provided in Appendix B.2. □

The proof follows Lemma 1. Theorem 1 states that backpropagating a vector of all ones ($\mathbf{1}_N$) is equivalent to backpropagating $N$ one-hot vectors with ones at orthogonal positions. This reduces the time complexity from $\mathcal{O}(N)$ to a constant $\mathcal{O}(1)$ since we only require a single pass to backpropagate $\mathbf{1}_N$. Hence, our introspective feature is given by,

$$r_x = \nabla_{W_L} J(\mathbf{1}_N, \hat{y}), r_x \in \Re^{d_{L-1} \times N}, \mathbf{1}_N = 1^{N \times 1} \tag{6}$$

Note the LHS is now $r_x$ instead of $r_I$ from Eq. 5. The final introspective feature is a matrix of the same size as $W_L$ extracted in $\mathcal{O}(1)$ with a space complexity of $\mathcal{O}(d_{L-1} \times N)$. $r_x$ is vectorized and scaled between $[-1, 1]$ before being used in Sections 5 and 6 as introspective features.

## 3  INTROSPECTIVE NETWORK

Once $r_x$ are extracted using Eq. 6, the introspective label $\tilde{y}$ from Fig. 1 is given by $\tilde{y} = \mathcal{H}(r_x)$. In this section, we analyze $\mathcal{H}(\cdot)$. From Fig. 1, $f(\cdot)$ is any existing trained network used to obtain introspective features $r_x$. It is trained to predict the ground truth $y$ given any $x$. Let $f(\cdot)$ be trained using the mean squared error loss function. Based on the assumption that $\mathcal{H}(r_x) = \mathbb{E}(y|f(x))$ and hence expectation of $y - \mathcal{H}(r_x)$ is 0, the loss function can be decomposed as,

$$\mathbb{E}[(f(x) - y)^2] = \mathbb{E}[(f(x) - \mathcal{H}(r_x))^2)] + \mathbb{E}[(\mathcal{H}(r_x) - y)^2)]. \tag{7}$$

Note that since the goal is to predict $y$ given $x$, $\mathcal{H}(r_x) = \mathbb{E}(y|f(x))$ is a fair assumption to make. Substituting for $f(x)$ in Eq. 7, and using variance decomposition of $y$ onto $f(x)$, we have,

$$\mathbb{E}[(\hat{y} - y)^2] = \text{Var}(\hat{y}) - \text{Var}(\mathcal{H}(r_x)) + \mathbb{E}[(\mathcal{H}(r_x) - y)^2]. \tag{8}$$

This decomposition is adopted from structured calibration techniques. A full derivation is presented in Kuleshov & Liang (2015). The first term $\text{Var}(\hat{y})$ is the the variance in the prediction from $f(\cdot)$. This term is the precision of $f(\cdot)$ and is low for a well trained network. The third term is the MSE function between the introspective network $\mathcal{H}(\cdot)$ and the ground truth. It is minimized while training the $\mathcal{H}(\cdot)$ network. The second term is the variance of the network $\mathcal{H}(\cdot)$ given features $r_x$. Note that minimizing Eq. 8 can occur by maximizing $\text{Var}(\mathcal{H}(r_x))$. However, $\text{Var}(\mathcal{H}(r_x))$ is also part of the bias-variance decomposition in the third term which is minimized. This prevents *perpetual introspection*, i.e having multiple gradient extraction-based $\mathcal{H}(\cdot)$ networks bootstrapped together, by creating a trade-off. We use a fisher vector interpretation to analyze $\text{Var}(\mathcal{H}(r_x))$. If $\mathcal{H}(\cdot)$ is a linear layer with parameters $W_{\mathcal{H}}$, the $\text{Var}(\mathcal{H}(r_x))$ term reduces to $W_{\mathcal{H}}^T W_{\mathcal{H}} \times \text{Var}(r_x) \propto \text{Tr}(r_x^T \Sigma^{-1} r_x)$ where $\Sigma$ is the covariance matrix. $\Sigma$ is a gaussian approximation for the shape of the manifold. Generalizing it to a higher dimensional manifold and replacing $\Sigma$ with $F$, we have,

$$\text{Var}(\mathcal{H}(r_x)) = \text{Tr}(r_x^T F^{-1} r_x), \tag{9}$$

$$\text{Var}(\mathcal{H}(r_x)) = \sum_{j=1}^{N} r_j^T F^{-1} r_j. \tag{10}$$

The RHS of Eq. 10 is a sum of fisher vectors taken across all possible labels. We analyze two cases of usage of $\mathcal{H}(\cdot)$ through Fisher Vectors : When input $\mathcal{X}$ is same as the training distribution and when $\mathcal{X}'$ is from a noisy distribution.

**Estimation of $\mathcal{X}$ using $\mathcal{H}$** When a sample $x \in \mathcal{X}$ is provided to a network $f(\cdot)$ trained on $\mathcal{X}$, all $r_j, j \neq \hat{y}$ in Eq. 10 tend to 0. The RHS reduces to $r_{\hat{y}}^T F^{-1} r_{\hat{y}}$. $r_{\hat{y}}$ is a function of $f(x)$ only and hence adds no new information to the framework. The results of $\mathcal{H}(\cdot)$ remain the same as $f(\cdot)$. In other words, given a trained ResNet-18 on CIFAR-10, the results of feed-forward learning will be the same as introspective learning on CIFAR-10 testset.

**Generalization to $\mathcal{X}'$ using $\mathcal{H}$** When a new samples $x' \notin \mathcal{X}$ is provided to a network $f(\cdot)$ trained on $\mathcal{X}$, a fisher vector based projection across labels is more descriptive compared to a feed-forward approach. The $N$ gradients in Eq. 10 add new information based on how the network needs to change the manifold shape $F$ to accomodate the introspective gradients. Hence, given a distorted version of CIFAR-10 testset, our proposed introspective learning generalizes with a higher accuracy while providing calibrated outputs from Eq. 8. We show these two claims in Section 5. The benefits of calibrated generalizability are further explored in downstream tasks like active learning and out-of-distribution detection in Section 6.

## 4 RELATED WORKS

**Two-stage Networks** The usage of two-stage approaches to inference in neural networks is not new. The authors in Chen et al. (2020b) propose SimCLR, a self-supervised framework where multiple data augmentation strategies are used to contrastively train an overhead MLP. The MLP provides features which are stored as a dictionary. This feature dictionary is used as a look-up table for new test data. The classical object detection technique of R-CNN (Girshick et al., 2014) uses separate feature extraction and detection stages for inference. In all these works, the extracted features are feed-forward activations. In this paper, we use gradients against all classes as features. Zhou & Levine (2021) and Bibas et al. (2019) consider all classes in a conditional maximum likelihood estimate on test data to retrain the model. These works differ from ours in our usage of the pairwise orthogonality of logits. We make use of this by having $\mathcal{H}(\cdot)$ as a classifier that explicitly learns the introspected pairwise relationships between classes.

**Gradients-as-Features** The gradients from a base network have been utilized in diverse applications including *post-hoc* visual explanations (Selvaraju et al., 2017; Prabhushankar et al., 2020), adversarial attacks (Goodfellow et al., 2014), and anomaly detection (Kwon et al., 2020) among others. In explanations, gradients are used to highlight features while in adversarial attacks, gradients characterize the required alterations to the features. Fisher Vectors use gradients of generative models to characterize the change that data creates within features (Jaakkola et al., 1999). A formulation similar to that of Fisher Kernels is used in Cohn (1994). Gradients of parameters are used to characterize the change in manifolds when new data is introduced to an already trained manifold. Our framework uses the intuition from Cohn (1994) to characterize changes for a datapoint that is perceived as new, due to it being assigned an introspective class that is different from its predicted class. In Zinkevich et al. (2017), the authors view the network as a graph and intervene within it to obtain *holographic* features. Our introspective features are also *holographic* in the sense that they are not true. However, our features are dependent on the notions from the network itself and do not require engineered interventions that can become expensive with scale. Mu et al. (2020) use gradients and activations together as features and note that the validity of gradients as features is in pretrained base networks rather than additional parameters from the two-stage networks. We demonstrate this as well in Appendix C.4.

**Augmentations and Robustness** The considered $r_x$ features from Eq. 6 can be considered as feature augmentations. Augmentations, including SimCLR, Augmix (Hendrycks et al., 2019), adversarial augmentation (Hendrycks & Dietterich, 2019), and noise augmentations (Vasiljevic et al., 2016) have shown to increase robustness of neural networks. We use introspection on top of non-augmented (Section 5) and augmented (Appendix C.2) networks and show that our proposed two-stage framework increases the robustness to create generalizable and calibrated inferences which aids active learning and out-of-distribution (OOD) detection. The same framework that robustly recognizes images despite noise can also detect noise to make an out-of-distribution detection.

**Confidence and Uncertainty** The existence of adversarial images (Goodfellow et al., 2014) heuristically decouples the probability of neural network predictions from confidence and uncertainty. A number of works including Sensoy et al. (2018) and MacKay (1995) use bayesian formulation to provide uncertainty. However, in downstream tasks like active learning and Out-Of-Distribution

(OOD) detection applications, existing state-of-the-art methods utilize softmax probability as confidences. This is because of the simplicity and ease of numerical computation of softmax. In active learning, uncertainty is quantified by the entropy (Wang & Shang, 2014), least confidence (Wang & Shang, 2014), or maximum margin (Roth & Small, 2006) of predicted logits, or through extracted features in BADGE Ash et al. (2019), and BALD (Gal et al., 2017). In OOD detection, Hendrycks & Gimpel (2016) propose Maximum Softmax Probability (MSP) as a baseline method by creating a threshold function on the softmax output. Liang et al. (2017) propose ODIN and improved on MSP by calibrating the network's softmax probability using temperature scaling (Guo et al., 2017). In this paper, we show that the proposed introspective features are better calibrated than their feedforward counterparts. Hence existing methods in active learning and OOD detection have a superior performance when using $\mathcal{H}(\cdot)$ to make predictions.

**Human Introspection**   There is no direct application that tests visual human introspection. In its absence, we choose the application of Full-Reference Image Quality Assessment (FR-IQA) to connect machine vision with human vision. The goal in FR-IQA is to objectively estimate the subjective quality of an image. Humans are shown a pristine image along with a distorted image and asked to score the quality of the distorted image (Sheikh et al., 2006). This requires reflection on the part of the observers. We take an existing algorithm (Temel et al., 2016) and show that introspecting on top of this IQA technique brings its assessed scores closer to human scores.

## 5   EXPERIMENTS

Across Sections 5 and 6, we use a 3-layered MLP with sigmoid activations as $\mathcal{H}$. The structure is presented in Appendix C.1. We first define generalization and calibration in the context of this paper.

**Generalization**   In this paper, without loss of consistency with related works, we say that the network trained on distribution $\mathcal{X}$ is generalizable if it predicts correctly on a shifted distribution $\mathcal{X}'$. The difference in data distributions can be because of data acquisition setups, environmental conditions, distortions among others. We use CIFAR-10 for $\mathcal{X}$ and two distortion datasets - CIFAR-10C (Hendrycks & Dietterich, 2019) and CIFAR-10-CURE (Temel et al., 2018) as $\mathcal{X}'$. Generalization is measured through performance accuracy.

**Calibration**   Given a data distribution $x \in \mathcal{X}$, belonging to any of $y \in [1, N]$, a neural network provides two outputs - the decision $\hat{y}$ and the confidence associated with $\hat{y}$, given by $\hat{p}$. Let $p$ be the true probability empirically estimated as $p = \hat{p}_i, \forall i \in [1, M]$. Then calibration is given by (Guo et al., 2017),

$$\mathbb{P}(y = \hat{y} | p = \hat{p}) = p \tag{11}$$

Calibration measures the difference between the confidence levels and the prediction accuracy. To showcase calibration we use the metric of Expected Calibration Error (ECE) as described in (Guo et al., 2017). The network predictions are placed in 10 separate bins based on their prediction confidences. Ideally, the accuracy equals the mid-point of confidence bins. The difference between accuracy and mid-point of bins, across bins is measured by ECE. Lower the ECE, better calibrated is the network.

**Datasets and networks**   CIFAR-10C consists of $950,000$ images whose purpose is to evaluate the robustness of networks trained on original CIFAR-10 trainset. CIFAR-10C perturbs the CIFAR-10 testset using 19 distortions in 5 progressive levels. Hence, there are 95 separate $\mathcal{X}'$ distributions to test on with each $\mathcal{X}'$ consisting of 10000 images. Note that we are not using any distortions or data from CIFAR-10C as a validation split during training. The authors in Temel et al. (2018) provide realistic distortions that they used to benchmark real-world recognition applications including Amazon Rekognition and Microsoft Azure. We use these distortions to perturb the test set of CIFAR-10. There are 6 distortions, each with 5 progressive levels. Of these 6 distortions - Salt and Pepper, Over Exposure, and Under Exposure noises are new compared to CIFAR-10C. We train four ResNet architectures - ResNet-18, 34, 50, and 101 He et al. (2016). All four ResNets are evaluated as sensing networks $f(\cdot)$. The training procedure and hyperparameters are presented in Appendix C.1.

**Testing on CIFAR-10 testset**   The trained networks are tested on CIFAR-10 testset with accuracies $91.02\%$, $93.01\%$, $93.09\%$, and $93.11\%$ respectively. Next we extract $r_x$ on all training and testing images in CIFAR-10. $\mathcal{H}(\cdot)$ is trained using $r_x$ from the trainset using the same procedure as $f(\cdot)$. When tested on $r_x$ of the testset, the accuracy for ResNets-18,34,50,101 is $90.93\%$, $92.92\%$, $93.17\%$,

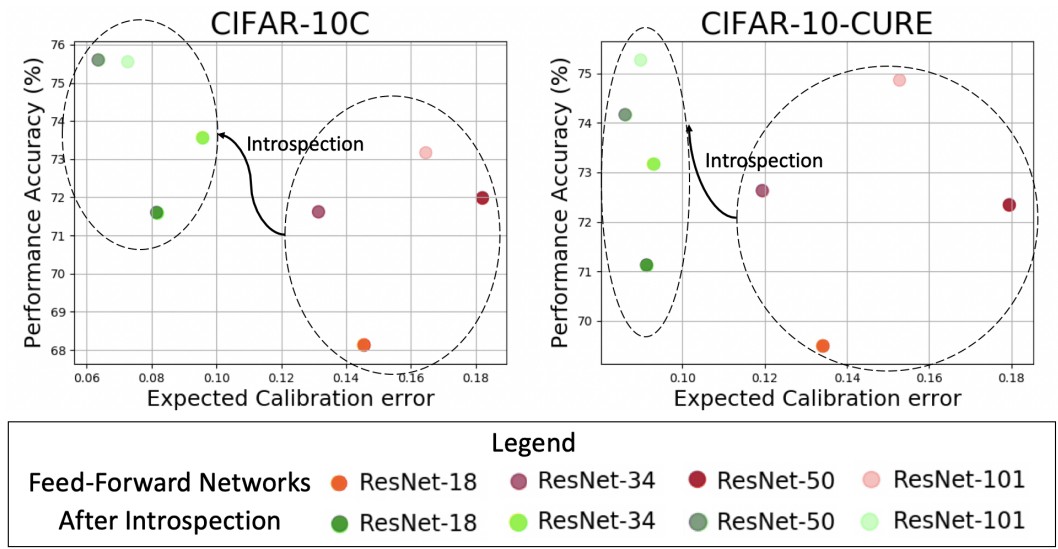

Figure 2: Scatter plot with performance accuracy vs expected calibration error. Ideally, networks are in top left. Introspectivity increases performance accuracy while decreasing calibration error.

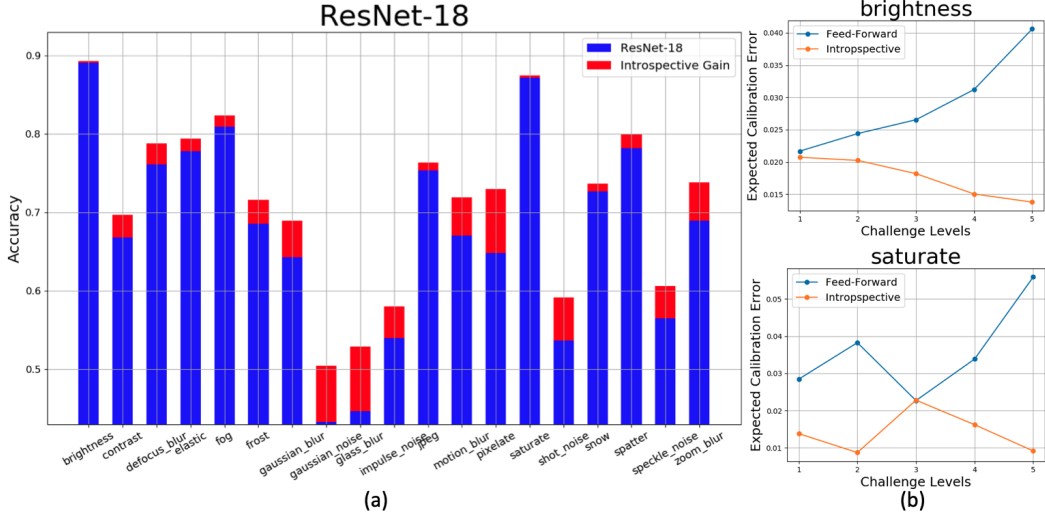

Figure 3: (a) ResNet-18 on CIFAR-10C. (b) Expected calibration error across 5 challenge levels in brightness and saturate distortions. Note that both these distortions do not affect the performance of the network and their feed-forward accuracy is high. The improvement in accuracy is statistically insignificant. However, introspection decreases the ECE across challenge levels.

and $93.03\%$. Note that this is similar to the feed-forward results. The average ECE of all feed-forward and introspective networks is $0.04$. Hence, when the test distribution is the same as training distribution there is no change in performance.

**Testing on CIFAR-10C and CIFAR-10-CURE** The results of all networks averaged across distortions in both the datasets are shown in Fig. 2. Note that in each case, there is a shift leftward and upward indicating that the performance improves while the calibration error decreases. In the larger CIFAR-10C dataset, the introspective ResNet-18 performs similar to ResNets-34 and 50 in terms of accuracy while beating them both in calibration. A more fine-grained analysis is shown in Fig. 3 for ResNet-18. The blue bars in Fig. 3a) represent the feed-forward accuracy. The red bars are the introspective accuracy gains over the feed-forward accuracy. Among 7 of the 19 distortions, the accuracy gains are over $5\%$. In Appendix C.2.1 and Fig. 6, we see that the gains are higher when the distortions are higher. Introspection performs well on blur-like distortions while struggling with distortions that disrupt the lower level characteristics of the image like brightness, contrast, and

Table 1: Recognition accuracy of Active Learning strategies.

| Methods | Architecture | Original Testset | | Gaussian Noise | |
|---|---|---|---|---|---|
| | | R-18 | R-34 | R-18 | R-34 |
| Entropy | Feed-Forward | 0.365 | 0.358 | 0.244 | 0.249 |
| | Introspective | 0.365 | 0.359 | **0.258** | **0.255** |
| Least Confidence | Feed-Forward | 0.371 | 0.359 | 0.252 | 0.25 |
| | Introspective | 0.373 | 0.362 | **0.264** | **0.26** |
| Margin | Feed-Forward | 0.38 | 0.369 | 0.251 | 0.253 |
| | Introspective | 0.381 | 0.373 | **0.265** | **0.263** |
| BALD | Feed-Forward | 0.393 | 0.368 | 0.26 | 0.253 |
| | Introspective | 0.396 | 0.375 | **0.273** | **0.263** |
| BADGE | Feed-Forward | 0.388 | 0.37 | 0.25 | 0.247 |
| | Introspective | 0.39 | 0.37 | **0.265** | **0.260** |

Table 2: OOD techniques applied on feed-forward and introspective networks when the data is under adversarial attack.

| Methods | OOD Datasets (Attack) | FPR (95% at TPR) ↓ | Detection Error ↓ | AUROC ↑ |
|---|---|---|---|---|
| | | Feed-Forward/Introspective | | |
| MSP | Textures | 99.98/**23.19** | 45.9/**7.9** | 30.4/**96.48** |
| | iSUN | 98.63/**87.2** | 46.71/**28.95** | 46.44/**75.81** |
| | Places365 | 100/**83.59** | 47.64/**26.46** | 25.08/**79** |
| | LSUN | 99.65/**87.64** | 43.38/**26.31** | 43.47/**78.4** |
| ODIN | Textures | 99.95/**2.06** | 47.7/**3.48** | 37.5/**99.11** |
| | iSUN | 96.8/**90.42** | 44.77/**31.11** | 53.88/**73.22** |
| | Places-365 | 99.97/**82.5** | 47.12/**26.86** | 32.69/**78.88** |
| | LSUN | 98.6/**88.28** | 40.51/**27.88** | 56.7/ **77.25** |

saturate. This can be attributed to the fact that $r_x$ are derived from the last layer of $f(\cdot)$ and are missing low-level statistics that are filtered out by network in the initial layers. However, in Fig. 3b), we show ECE for brightness and saturate distortions across all 5 distortion levels - higher the level, more is the distortion affecting $\mathcal{X}'$. It can be seen that while the ECE for feed-forward networks increases across levels, the ECE for introspective networks decrease. Hence, even when there are no accuracy gains to be had, introspection helps in calibration.

**Plug-in results of Introspection**   Note that there are a number of techniques proposed to alleviate a neural network's robustness challenges against distortions. The authors in (Vasiljevic et al., 2016) show that finetuning VGG-16 using blurry training images increases the performance of classification under blurry conditions. (Temel et al., 2017) propose utilizing distorted virtual images to boost performance accuracy. The authors in (Hendrycks & Dietterich, 2019) use adversarial images to augment the training data. All these works require knowledge of distortion or large amounts of new data during training. Our proposed method can infer introspectively on top of any existing $f(\cdot)$ enhanced using existing methods. In Appendix C.2, we show performance on top of (Vasiljevic et al., 2016) and (Hendrycks & Dietterich, 2019) of $6.8\%$ on Level 5 distortions. In Appendix C.3, we analyze SimCLR and show that introspecting on the self supervised features increases its CIFAR-10C performance by about $6\%$ on ResNet-101. We introspect on top of Augmix (Hendrycks et al., 2019) and show that while recognition accuracy is the same, introspection reduces ECE of Augmix network by $43.33\%$. A number of ablation studies including analysis of structure of $\mathcal{H}$, loss function, distortion levels on performance accuracy and ECE are shown in Appendix C.4. Moreover, we examine introspection when $\mathcal{X}'$ is domain shifted data from Office (Saenko et al., 2010) dataset in Appendix C.6.

# 6   APPLICATIONS

In this section, we illustrate the advantages of introspective networks in two applications that are a function of both generalization and calibration - active learning and out-of-distribution detection. In both these applications, we show that the existing state-of-the-art methods perform better in their respective tasks and metrics if they were applied on $\mathcal{H}(\cdot)$ than on $f(\cdot)$. We then conclude by grounding the proposed introspection in neural nets with introspection in humans through IQA.

**Active Learning**   The goal in active learning is to decrease the test error in a model by choosing the *best* samples from a large pool of unlabeled data to annotate and train the model. A number of strategies are proposed to query the *best* samples. A full review of active learning and query strategies are given in Settles (2009). Existing active learning strategies define *best* samples to annotate as those samples that the model is most uncertain about. We use the strategies given in Section 4 to showcase the effectiveness of $\mathcal{H}(\cdot)$. We show the results of ResNet-18 and 34 architecture in Table 1. Implementations of all query strategies in Table 1 are taken from the codebase of Ash et al. (2019) and reported as feed-forward results. Note that the query strategies act on $f(\cdot)$ to sample images at every round. In the introspective results, all query strategies sample using $\mathcal{H}(\cdot)$. The training and testing procedures strategies are the same as feed-forward from Ash et al. (2019). Doing so we find similar results as recognition - on the original testset the active learning results are the same while there is a gain across strategies on Gaussian noise testset from CIFAR-10C. Note that the results shown are

averaged over 20 rounds with a query batch size of a 1000 and initial random choice - which were kept same for $f(\cdot)$ and $\mathcal{H}(\cdot)$ - of 100. Further details and plots are shown in Appendix C.8.

**Out-of-distribution Detection**    The goal of Out-Of-Distribution (OOD) detection is to detect those samples that are drawn from a distribution $\mathcal{X}' \neq \mathcal{X}$ given a fully trained $f(\cdot)$. As mentioned in Section 4, we use MSP (Hendrycks & Gimpel, 2016) and ODIN (Liang et al., 2017) to illustrate the effectiveness of existing OOD methods when applied on $\mathcal{H}(\cdot)$ than if they were applied on the feed-forward $f(\cdot)$. The code for OOD detection techniques are taken from Chen et al. (2020a) along with all hyperparameters and the training regimen. The temperature scaling coefficient for ODIN is set to 1000. Note that we do not use additional temperature scaling on $\mathcal{H}(\cdot)$ to illustrate the effectiveness of our method. We use three established metrics to evaluate OOD detection - False Positive Rate (FPR) at 95% True Positive Rate (TPR), Detection error, and AUROC. Ideally, AUROC values for a given method is high while the other two metrics are low. We use CIFAR-10 as our in-distribution dataset and use four OOD datasets - iSUN (Xiao et al., 2010), Describable Textures Dataset (Cimpoi et al., 2014), Places 365 (Zhou et al., 2017), and LSUN (Yu et al., 2015). As in recognition and active learning, we consider two difficulty settings. The first is the vanilla case with the above-listed out-of-distribution datasets. The results are presented in Appendix C.9 and Table 11. The harder setting is when the datasets are all attacked with adversarial noise on top of being OOD. This setting is formalized in Chen et al. (2020a). We show these results in Table 2. In all cases in Table 2, introspection outperforms its feed-forward counterpart. In the vanilla setting, introspection is better than its feed-forward network in 15 of considered 24 metrics. This is inline with results from recognition and active learning where introspection generalizes better on $\mathcal{X}'$ compared to $\mathcal{X}$ testsets.

**Image Quality Assessment (IQA)**    TID 2013 (Ponomarenko et al., 2015) and MULTI-LIVE (Jayaraman et al., 2012) are two IQA datasets with 3000 and 225 distorted images respectively in 5 and 4 progressively increasing levels of distortions. This setup is similar to CIFAR-10C. The goal is to objectively assess the subjective quality of the distorted images given the pristine image. Inline with existing techniques (Temel et al., 2016), we use five metrics to measure the similarity between the algorithmically predicted qualities and human qualities - outlier ratio (consistency), root mean square error (RMSE, accuracy), Pearson correlation (PRCC, linearity), Spearman correlation (SRCC, rank), and Kendall correlation (KRCC, rank). To do so, we use an existing IQA technique called UNIQUE (Temel et al., 2016) and introspect on top of it. The details of UNIQUE as well as related works and compared methods are presented in Appendix C.10. Since UNIQUE is an autoencoder architecture and since the pristine image is available, introspection can occur based on features and on earlier layers. The exact procedure is given in Appendix C.10 and is termed as Introspective-UNIQUE. The results are shown in Table 12. The proposed framework acts as a plug-in on top of UNIQUE. For instance, UNIQUE is the third best performing method in MULTI dataset in terms of RMSE, PLCC, SRCC, and KRCC. However, Introspective-UNIQUE improves the performance for these metrics by 1.315, 0.036, 0.020, and 0.023, respectively and achieves the best performance on all metrics.

## 7  DISCUSSION AND CONCLUSION

**Limitations and future work**    The paper illustrates the benefits of utilizing the change in model parameters as a measure of model introspection. In Section 2.2, we analyze the time complexity and accelerate it to $\mathcal{O}(1)$. However, the space complexity is still dependent on $N$. The paper uses an MLP for $\mathcal{H}(\cdot)$ and constructs $r_x$ by vectorizing extracted gradients. Hence, taking the dimensionality of the final feature layer from Eq. 1, the space complexity is $\mathcal{O}(N \times d^{L-1})$. For large datasets with large $N$, usage of $r_x$ as a vector of concatenated gradients is prohibitive. Hence, a required future work is to provide a method of combining all $N$ gradients without vectorization. Also, our implementation uses serial gradient extraction across images. This is non-ideal since the available GPU resources are not fully utilized. A parallel implementation with per-sample gradient extraction (Goodfellow, 2015) is a pertinent acceleration technique for the future.

**Conclusion**    We introduce the concept of introspection in neural networks as two separate stages in a network's decision process - the first is making a quick assessment based on sensed patterns in data and the second is reflecting on that assessment based on all possible decisions that could have been taken and making a final decision based on this reflection. We show that doing so increases the generalization performance of neural networks as measured against distributionally shifted data while reducing the calibration error of neural networks. Existing state-of-the-art methods in downstream tasks like active learning and out-of-distribution detection perform better in an introspective setting compared to a feed-forward setting especially when the distributional difference is high.

## 8 ETHICS STATEMENT

The introspective explanations can serve to examine the intrinsic notions and biases that a network uses to categorize data since $\mathcal{H}(\cdot)$ obtains its introspective answers through $f(\cdot)$. However, any internal bias present in $f(\cdot)$ only gets strengthened in $\mathcal{H}(\cdot)$ through confirmation bias. The framework will benefit from a human intervention between $f(\cdot)$ and $\mathcal{H}(\cdot)$ in sensitive applications. One way would be to ask counterfactual questions by providing an established counterfactual and asking the network to reflect based on that. While the introspective framework will remain the same, the features will change. Such a confirmation bias is also present in humans. In his seminal book in 2011, Kahneman (2011) outlines two systems of thought and reasoning in humans - a fast and instinctive 'system 1' that heuristically associates sensed patterns followed by a more deliberate and slower 'system 2' that examines and analyzes the data in context of intrinsic notions. Our framework derives its intuition based on these two systems of reasoning.

## 9 REPRODUCIBILITY STATEMENT

The paper uses publicly available datasets to showcase the results. Our introspective learning framework is built on top of existing deep learning apparatus - including ResNet architectures (He et al., 2016) (inbuilt PyTorch architectures), CIFAR-10C data (Hendrycks & Dietterich, 2019) (source code at `https://zenodo.org/record/2535967#.YLpTF-1KhhE`), calibration ECE and MCE metrics (Guo et al., 2017) (source code at `https://github.com/markus93/NN_calibration`), out-of-distribution detection metrics, and codes for existing methods were adapted from (Chen et al., 2020a) (source code at `https://github.com/jfc43/robust-ood-detection`), active learning methods and their codes were adapted from (Ash et al., 2019) (source code at `https://github.com/JordanAsh/badge`), Grad-CAM was adapted from (Selvaraju et al., 2017) (code used is at `https://github.com/adityac94/Grad_CAM_plus_plus`). Our own codes will be released upon acceptance. The exact training hyperparameters for $f(\cdot)$ and $\mathcal{H}(\cdot)$, and all considered $\mathcal{H}(\cdot)$ architectures are shown in Appendix C.1. Extensive ablation studies on $\mathcal{H}(\cdot)$ are shown in C.4.

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

## A   Appendix : Introspection, Reasoning, and Explanations

Introspection was formalized by Wundt (1874) as a field in psychology to understand the concepts of memory, feeling, and volition (Schwitzgebel, 2019). The primary focus of introspection is in reflecting on oneself through directed questions. While the directed questions are an open field of study in psychology, we use reasoning as a means of questions in this paper. Specifically, abductive reasoning. Abductive reasoning was introduced by the philosopher Charles Sanders Peirce (Peirce, 1931), who saw abduction as a reasoning process from effect to cause (Paul, 1993). An abductive reasoning framework creates a hypothesis and tests its validity without considering the cause. From the perspective of introspection, a hypothesis can be considered as an answer to one of the three following questions: a causal *'Why P?'* question, a counterfactual *'What if?'* question, and a contrastive *'Why P, rather than Q?'* question. Here $P$ is the prediction and $Q$ is any contrast class. Both the causal and counterfactual questions require active interventions for answers. These questions

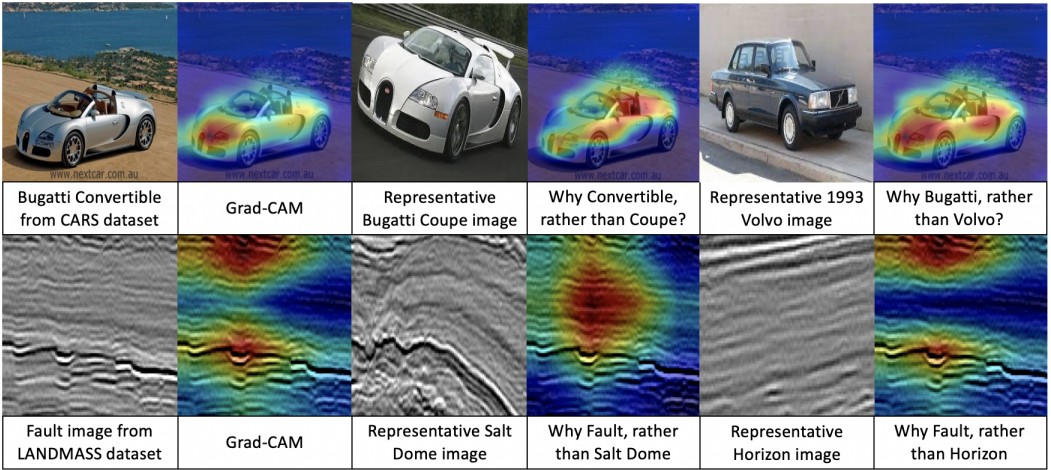

Figure 4: Introspective feature visualizations. The images in the leftmost column are the input $x$. The representative images are for illustrative purposes and are not used to extract features.

try to assess the causality of some endogenous or exogenous variable. However, introspection is the assessment of ones own notions rather than an external variable. Hence, a contrastive question of the form *'Why P, rather than Q?'* lends itself as the directed question for introspection. Here $Q$ is the introspective class. It has the additional advantage that the network $f(\cdot)$ serves as the knowledge base of notions. All reflection images from 1, Fig. 4, and Fig. 5 are contrastive. We describe the generation process of these *post-hoc* explanations.

**Introspective Feature Visualization** We modify Grad-CAM (Selvaraju et al., 2017) to visualize $r_j$ from Eq. 2. Grad-CAM visually justifies the decision made by $f(\cdot)$ by highlighting features that lead to $\hat{y}$. It does so by backpropagating the logit associated with the prediction, $\hat{y}$. The resulting gradients at every feature map are global average pooled and used as importance scores. The importance scores multiply the activations of the final convolutional layer and the resultant map is the Grad-CAM visualization. Hence, gradients highlight the activation areas that maximally lead to the prediction $\hat{y}$. In Fig. 1, given a spoonbill image $x$ and a ImageNet-pretrained (Deng et al., 2009) VGG-16 network, the sensing visualization shown is Grad-CAM. Grad-CAM indicates that the pink and round body, and straight beak are the reasons for the decision. Instead of backpropagating the $\hat{y}$ logit, we backpropagate $J(y_I, \hat{y})$ in the Grad-CAM framework. The gradients represent introspective features and are used as importance scores. It can be seen that they visually highlight the explanations to *'Why $\hat{y}$, rather than $y_I$'*. In Fig. 1, the network highlights the neck of the spoonbill to indicate that since an S-shaped neck is not observed, $x$ cannot be a flamingo. Similarly, the body of the spoonbill is highlighted when asked why $x$ is not a crane since cranes have white feathers while spoonbills are pink. Two more examples are shown in Fig. 4. In the first row, a VGG-16 architecture is trained on Stanford Cars dataset (Krause et al., 2013). Given a Bugatti convertible image, Grad-CAM highlights the bonnet as the classifying factor. An introspective question of why it cannot be a bugatti coupe is answered by highlighting the open top of the convertible. The entire car is highlighted to differentiate the bugatti convertible from a Volvo. In the second row, we explore visual explanations in computed seismic images using LANDMASS dataset (Alaudah et al., 2018). A ResNet-18 architecture using the procedure from Shafiq et al. (2018) is trained. The dataset has four geological features as classes - faults, salt domes, horizons, and chaotic regions. Given a fault image in Fig. 4, Grad-CAM highlights the regions where the faults are clearly visible as fractures between rocks. However, these regions resemble salt domes as shown in the representative image. The introspective answer of why $x$ is not predicted as a salt dome tracks a fault instead of highlighting a general region that also resembles a salt dome. Note that no representative images are required to obtain introspective visualizations. The gradients introspect based on notions of classes in network parameters.

# B  APPENDIX : PROOFS

## B.1  PROOF FOR LEMMA 1

We start by assuming $J(\cdot)$ is a cross-entropy loss. $J(y_I, \hat{y}), I \in [1, N]$ can also be written as,

$$J(y_I, \hat{y}) = -y_{\hat{y}} + \log \sum_{j=1}^{N} e^{y_j}, \text{ where } \hat{y} = f(x), \hat{y} \in \Re^{N \times 1}. \tag{12}$$

This definition is used in PyTorch to implement cross entropy. Here we assume that the predicted logit, i.e, the argument of the max value in the logits $\hat{y}$ is $y_{\hat{y}}$. While training, $y_{\hat{y}}$ is the true label. In this paper, we backpropagate any trained class $I$, as an introspective class. Hence, Eq. 12 can be rewritten as,

$$J(y_I, \hat{y}) = -y_I + \log \sum_{j=1}^{N} e^{y_j}, \text{ where } \hat{y} = f(x), \hat{y} \in \Re^{N \times 1}. \tag{13}$$

Approximating the exponent within the summation with its second order Taylor series expansion, we have,

$$J(y_I, \hat{y}) = -y_I + \log \sum_{j=1}^{N} \left(1 + y_j + \frac{y_j^2}{2}\right). \tag{14}$$

Note that for a well trained network $f()$, the logits of all but the predicted class are high. As noted before, the predicted logit is $y_{\hat{y}}$. Hence $\sum_{j=1}^{N} \frac{y_j^2}{2} = \frac{y_{\hat{y}}^2}{2}$. Substituting,

$$J(y_I, \hat{y}) = -y_I + \log(N) + \log(\sum_{j=1}^{N} y_j) + \log\left(\frac{y_{\hat{y}}^2}{2}\right). \tag{15}$$

The quantity in Eq. 15 is differentiated, hence nulling the effect of constant $\log(N)$. For a well trained network $f(\cdot)$, small changes in $W$ do not adversely affect the sum of all logits $\sum_{j=1}^{N} y_j$. Hence approximating its gradient to 0 and discarding it, we can obtain $\nabla_W J(y_j, \hat{y})$ as a function of two logits, $y_I$ and $y_{\hat{y}}$ given by,

$$\nabla_W J(y_I, \hat{y}) = -\nabla_W y_I + \nabla_W \log\left(\frac{y_{\hat{y}}^2}{2}\right). \tag{16}$$

$y_I$ is a one-hot vector of dimensionality $N \times 1$ while $\nabla_W$ is a $d_{L-1} \times N$ matrix. The product extracts only the $I^{th}$ filter in the $W$ matrix in gradient calculations. Following the above logic for $y_{\hat{y}}$, we have,

$$\nabla_W J(y_I, \hat{y}) = -\nabla_{W,I} y_I + \nabla_{W,y_{\hat{y}}} g(y_{\hat{y}}), \tag{17}$$

where $g(\cdot)$ is some function of $y_{\hat{y}}$. Hence the gradient $r_I = \nabla_W J(y_I, \hat{y})$ lies in the span of the filter gradients of $W_I$ and $W_{\hat{y}}$, making $r_I$ orthogonal to all other filter gradient pairs. Hence proven.

We demonstrate this sparsity in Fig. 5. A two-layer CNN is trained on MNIST (LeCun et al., 1998) dataset. MNIST is a handwritten digits dataset consisting of $50,000$ training images and $10,000$ testing images among 10 classes. Our two-layer CNN recognizes these digits with an averaged test accuracy exceeding $99\%$. The final fully connected layer in this network has a size of $50 \times 10$. Hence, the dimensionality of each filter $W_{L,i}, i \in [1, 10]$ that corresponds to a class is $50 \times 1$. We provide an input image $x$ of number 5 to a trained network as shown in Fig. 5. The network correctly identifies the image as a 5. We then backpropagate the introspective class 0 using the cross entropy loss $J(5, 0)$ with $P = 5$ and $Q = 0$. This answers the question *'Why 5, rather than 0?'*. The gradient features in the final fully connected layer are the same dimensions as the final fully connected layer - $50 \times 10$. This matrix is displayed as a normalized image in Fig. 5. Yellow scales to 1 and blue is $-1$ while green is 0. It can be seen that the only values present in the matrix are negative at $W_{L,0}$, in blue, and positive in $W_{L,5}$, in yellow. This validates Eq. 5 that for a fully-trained network the only values, and hence the only information, required from $W_L$ for $Q = 0$ is $\nabla_{W_{L,0}}$. We show the matrix $\nabla_{W_L}$ when $Q = 1, 2, 4, 5, 6$. The difference among all matrices is the location of the negative values that exist at $\nabla_{W_{L,Q}}$ for different values of $Q$.

Hence, for $N$ introspective features in 5, the space complexity of $r_x$ which is a concatenation of $N$ separate $r_i$, reduces from $\mathcal{O}(d_{L-1} \times N^2)$ to $\mathcal{O}(d_{L-1} \times N)$.

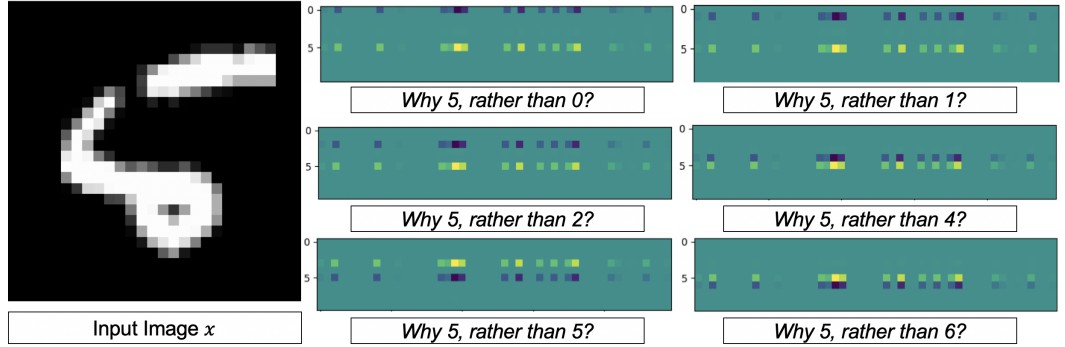

Figure 5: For the input image on the left, the $\nabla_{W_L} J(y_I, 5)$ are shown on the right. Each image is a visualization of the $50 \times 10$ gradient matrix. All images are sparse except in the prediction row 5 and introspective question row $i$.

### B.2 PROOF FOR THEOREM 1

The proof for Theorem 1 follows from Lemma 1. For any given data $x$, there are $N$ possible introspections and hence $N$ possible reflections. The LHS in Eq. 17 is summed across $N$ losses. Since $y_j, j \in [1, N]$ are one-hot vectors, they are orthogonal and the first term in RHS is an addition across $j$. The second term in RHS is independent of $j$. Representing this in equation form, we have,

$$\sum_{j=1}^{N} \nabla_W J(y_j, \hat{y}) = -\sum_{j=1}^{N} \nabla_{W,j} y_j + N \times \nabla_{W,y_{\hat{y}}} g(y_{\hat{y}}). \tag{18}$$

The first term is added $N$ times for $N$ orthogonal $y_I$. Hence, the first term reduces to a sum of all gradients of $j^{th}$ filters when backpropagating $y_j$. Removing the summation and replacing $y_j = \mathbf{1}_N$ or a vector of all ones in the LHS, we still have the same RHS given by,

$$\nabla_W J(\mathbf{1}_N, \hat{y}) = -\sum_{j=1}^{N} \nabla_{W,j} y_j + N \times \nabla_{W,y_{\hat{y}}} g(y_{\hat{y}}). \tag{19}$$

Equating the LHS from Eq. 18 and Eq. 19, we have the proof.

### B.3 TRADEOFF IN EQ. 8

Eq. 8 suggests a trade-off between minimizing $\mathbb{E}[(\mathcal{H}(r_x) - y)^2]$, which is the cost function for training $\mathcal{H}(\cdot)$, and the variance of the network $\mathcal{H}(\cdot)$. Ideally, an optimal point exists that optimally minimizes the cost function of $\mathcal{H}(\cdot)$ while maximizing its variance. This also prevents decomposing $\mathcal{H}(\cdot)$ into $\mathcal{H}_1(\cdot)$ and $\mathcal{H}_2(\cdot)$ that further introspect on $\mathcal{H}(\dot)$. In this paper, we create a single introspective network $\mathcal{H}(\cdot)$. Hence, we do not comment further on the practical nature of the trade-off or perpetual introspection. It is currently beyond the scope of this work. In all experiments, we train $\mathcal{H}(\cdot)$ as any other network feed-forward network - by minimizing an empirical loss function given the ground truth.

### B.4 FISHER VECTOR INTERPRETATION

We make two claims before Eq. 10 both of which are well established. These include :

- **Variance of a linear function** For a linear function $y = W \times x + b$, the variance of $y$ is given by $\text{Var}(Wx + b) = W^2 \text{Var}(x)$ if $\text{Var}(W) = 0$.
- **Variance of a linear function when $W$ is estimated by gradient descent** Ignoring the bias $b$, and taking $y = Wx = x^T \Sigma^{-1} x^T (xW)$, we have $\text{Var}(Wx) = \sigma^2 \text{Tr}(x^T \Sigma^{-1} x)$.

Both these results lead to Eq. 9. Since $r_x \in \Re^{d_{L-1} \times N}$, the trace of the matrix given by $\text{Tr}(r_x^T F^{-1} r_x)$, is a sum of projections on individual weight gradients given by $\sum_{j=1}^{N} r_j^T F^{-1} r_j$ in the Fisher sense.

Table 3: Structure of $\mathcal{H}(\cdot)$ and accuracies on CIFAR-10C as reported in the paper.

| $f(\cdot)$ | Part 1: Structure of $\mathcal{H}(\cdot)$ - All layers separated by sigmoid | Accuracy (%) |
|---|---|---|
| R-18,34 | $640 \times 300 - 300 \times 100 - 100 \times 10$ | 71.4, 73.36 |
| R-50, 101 | $2560 \times 300 - 300 \times 100 - 100 \times 10$ | 75.2, 75.47 |

# C  APPENDIX : ADDITIONAL RESULTS

## C.1  STRUCTURE OF $\mathcal{H}(\cdot)$ AND TRAINING DETAILS

In this section, we provide the structure of the proposed $\mathcal{H}(\cdot)$ architecture. Note that, from Eq. 1, the $y_{feat}$ in feed-forward learning are processed through a linear layer. We process the introspective features $r_x$ through an MLP $\mathcal{H}(\cdot)$, whose parameter structure is given in Table 3. Hence, we follow the same workflow as feed-forward networks in introspective learning. The feed-forward features $f_{L-1}(x)$ are passed through the last linear layer in $f(\cdot)$ to obtain the prediction $\hat{y}$. The introspective features are passed through an MLP to obtain the prediction $\tilde{y}$. The exact training procedure for $\mathcal{H}(\cdot)$ is presented below.

**Training $f(\cdot)$ and Hyperparameters**  We train four ResNet architectures - ResNet-18, 34, 50, and 101 He et al. (2016). Note that we are not using any known techniques that promote either generalization (training on noisy data (Vasiljevic et al., 2016)) or calibration (Temperature scaling (Guo et al., 2017)). The networks are trained from scratch on CIFAR-10 dataset which consists of $50000$ training images with 10 classes. The networks are trained for 200 epochs using SGD optimizer with momentum $= 0.9$ and weight decay $= 5e - 4$. The learning rate starts at $0.1$ and is changed as $0.02, 0.004, 0.0008$ after epochs $60, 120,$ and $160$ respectively. PyTorch in-built Random Horizontal Flip and standard CIFAR-10 normalization is used as preprocessing transforms.

**Training $\mathcal{H}(\cdot)$**  The structures of all MLPs are shown in Table 3. ResNet-18,34 trained on CIFAR-10 provide $r_x$ of dimensionality $640 \times 1$. This is fed into $\mathcal{H}(\cdot)$ which is trained to produce a $10 \times 1$ output. Note that $r_x$ from ResNet-50,101 are of dimensionality $2560 \times 1$ - due to larger dimension of $f_{L-1}()$. All MLPs are trained similar to $f(\cdot)$ - for 200 epochs, SGD optimizer, momentum $= 0.9$, weight decay $= 5e^{-3}$, learning rates of $0.1, 0.02, 0.004, 0.0008$ in epochs $1 - 60, 61 - 120, 121 - 160, 161 - 200$ respectively. For the larger 5-layered ResNet-50,101 networks in Table 8, dropout with $0.1$ is used and the weight decay is reduced to $5e^{-4}$.

## C.2  INTROSPECTIVE ACCURACY GAIN AND CALIBRATION ERROR STUDIES

In this section, we present additional recognition and calibration results. In Fig. 3a), we showed distortion-wise accuracy and the introspective gain for ResNet-18. In this section, we present level-wise and network-wise accuracies for all four considered ResNet architectures. We show that an introspective ResNet-18 matches a Feed-Forward ResNet-50 in terms of recognition performance. We then compare the results of ResNet-18 against existing techniques that promote robustness. We show that introspection is a plug-in approach that acts on top of existing methods and provides gain. We do the same for calibration experiments on CIFAR-10C where we provide level-wise distortion-wise graphs for Expected Calibration Error (ECE) similar to Fig. 3b).

### C.2.1  LEVEL-WISE RECOGNITION ON CIFAR-10C

In Fig. 6b), the introspective performance gains for the four networks are categorized based on the distortion levels. All 19 categories of distortion on CIFAR-10C are averaged for each level and their respective feed-forward accuracy and introspective gains are shown. Note that the levels are progressively more distorted. Hence, level 1 distribution $\mathcal{X}'$ is similar to the training distribution $\mathcal{X}$ when compared to level 5 distributions. As the distortion level increases, the introspective gains also increase. This is similar to the results from Section 6. In both active learning and OOD applications as $\mathcal{X}'$ deviates from $\mathcal{X}$, introspection performs better. In Fig. 6a), we show the distortion-wise and

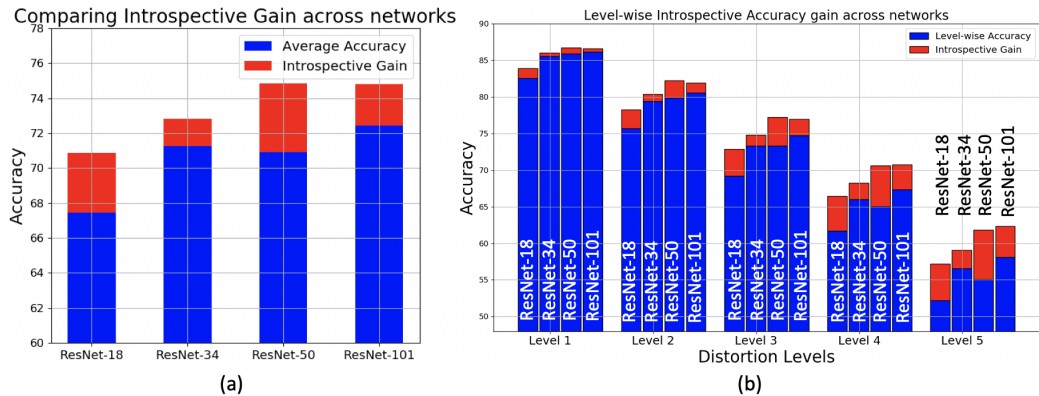

Figure 6: Introspective performance gains over Feed-Forward networks of a) ResNets-18,34,50,101, b) Level-wise averaged results across ResNets-18,34,50,101

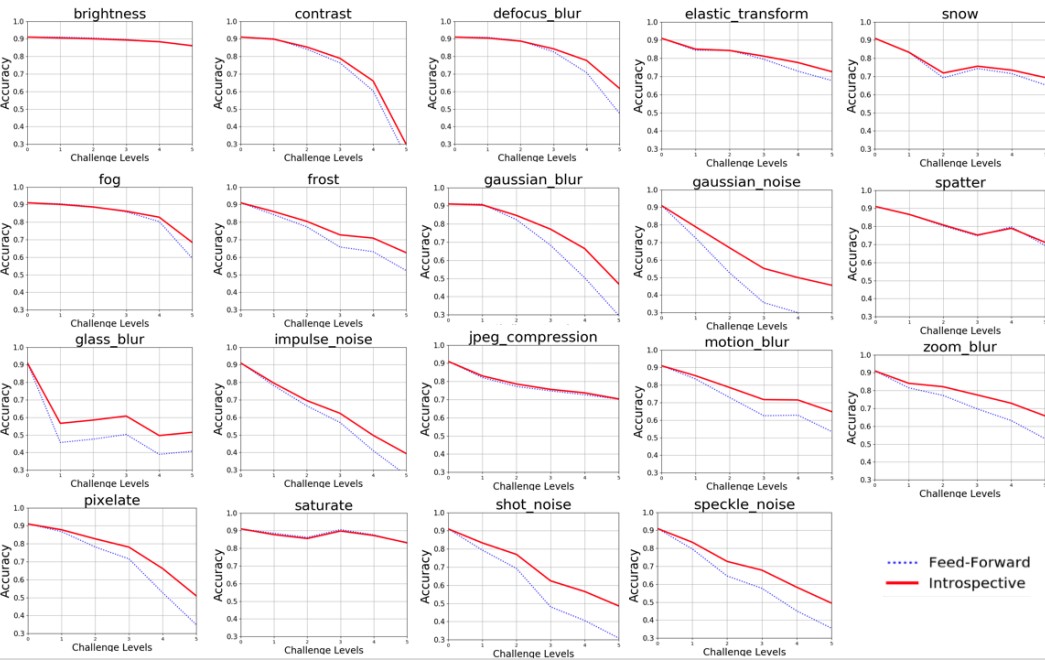

Figure 7: Introspective performance gains over Feed-Forward Resnet-18 across distortions and levels

level-wise increase for each network. Note that, an Introspective ResNet-18 performs similarly to a Feed-Forward ResNet-50.

### C.2.2 DISTORTION-WISE AND LEVEL-WISE RECOGNITION ON CIFAR-10C

In Fig. 7, the introspective accuracy performance for Resnet-18 across 19 distortions and 5 distortion levels is shown. Note that CIFAR-10C consists of 950,000 test images. The 4% increase in performance translates to around 35,000 more images correctly classified over its feed-forward counterpart. These gains are especially visible among Level 5 distortions.

### C.2.3 INTROSPECTION AS A PLUG-IN ON TOP OF EXISTING TECHNIQUES

Several techniques exist that boost the robustness of neural networks to distortions. These include training with noisy images (Vasiljevic et al., 2016), training with adversarial images (Hendrycks & Dietterich, 2019), and self-supervised methods like SimCLR (Chen et al., 2020b) that train by augmenting distortions. Another commonly used technique is to pre-process the noisy images to denoise them. All these techniques can be used to train $f(\cdot)$. Our proposed framework sits on top of

Table 4: Introspecting on top of existing robustness techniques.

| Methods | Accuracy |
|---|---|
| ResNet-18 | 67.89% |
| Denoising | 65.02% |
| Adversarial Train (Hendrycks & Dietterich, 2019) | 68.02% |
| SimCLR (Chen et al., 2020b) | 70.28% |
| Augment Noise (Vasiljevic et al., 2016) | 76.86% |
| Augmix (Hendrycks et al., 2019) | 89.85% |
| ResNet-18 + Introspection | 71.4% |
| Denoising + Introspection | 68.86% |
| Adversarial + Introspection | 70.86% |
| SimCLR + Introspection | 73.32% |
| Augment Noise + Introspection | 77.98% |
| Augmix + Introspection | 89.89% (ECE 43.33% ↓) |

any $f(\cdot)$. Hence, it can be used as a plug-in network. These results are shown in Table 4. Denoising 19 distortions is not a viable strategy assuming that the characteristics of the distortions are unknown. We use Non-Local Means denoising and the results obtained are lower than the feed-forward accuracy by almost $3\%$. However, introspecting on this model increases the results by $3.84\%$. We create untargeted adversarial images using I-FGSM attack with $\alpha = 0.01$ and use them to train a ResNet-18 architecture. In our experiments this did not increase the feed-forward accuracy. Introspecting on this network provides a gain of $2.84\%$. SimCLR Chen et al. (2020b) and introspection on SimCLR is discussed in Section C.3. In the final experimental setup of augmenting noise (Vasiljevic et al., 2016), we augment the training data of CIFAR-10 with six distortions - gaussian blur, salt and pepper, gaussian noise, overexposure, motion blur, and underexposure - to train a ResNet-18 network $f'(\cdot)$. We use the noise characteristics provided by (Temel et al., 2018) to randomly distort 500 CIFAR-10 training images by each of the six distortions. The original training set is augmented with the noisy data and trained. The results of the feed-forward $f'(\cdot)$ show a substantial increase in performance to $76.86\%$. This is about $9\%$ increase from the original architecture. We show that introspecting on $f'(\cdot)$ provides a further gain in accuracy of $1.12\%$. Note that to train $\mathcal{H}(\cdot)$, we do not use the augmented data. We only use the original CIFAR-10 undistorted training set. The gain obtained is by introspecting on only the undistorted data, even though $f'(\cdot)$ contains knowledge of the distorted data. Hence, introspection is a plug-in approach that works on top of any network $f(\cdot)$ or enhanced network $f'(\cdot)$. Augmix (Hendrycks et al., 2019) is currently the best performing technique on CIFAR-10C. It creates multiple chains of augmentations to train the base WideResNet network. On CIFAR-10C, $f'(\cdot)$ obtains $89.85\%$ recognition accuracy. We use $f'(\cdot)$ as our base sensing model and train an introspective MLP on $f'(\cdot)$. Note that we do not use any augmentations for training $\mathcal{H}(\cdot)$. Doing so, we obtain a statistically similar accuracy performance of $89.89\%$. However, the expected calibration error of the feed-forward $f'(\cdot)$ model decreases by $43.33\%$ after introspection. Hence, when there is no accuracy gains to be had, introspection provides calibrated models.

Table 5: Expected Calibration Error and Maximum Calibrated Error for Feed-Forward vs Introspective Networks.

| Architectures | | ResNet-18 | ResNet-34 | ResNet-50 | ResNet-101 |
|---|---|---|---|---|---|
| ECE (↓) | $f(\cdot)$ | 0.14 | 0.18 | 0.13 | 0.16 |
| | $\mathcal{H}(\cdot)$ | **0.07** | **0.09** | **0.06** | **0.1** |
| MCE (↓) | $f(\cdot)$ | 0.27 | 0.34 | 0.27 | 0.32 |
| | $\mathcal{H}(\cdot)$ | **0.23** | **0.24** | **0.25** | **0.23** |
| Brier Loss (↓) | $f(\cdot)$ | **0.046** | **0.045** | **0.041** | **0.042** |
| | $\mathcal{H}(\cdot)$ | 0.054 | 0.053 | 0.053 | 0.052 |

### C.2.4    EXPECTED CALIBRATION ERROR (ECE)

In Fig. 3b), we show ECE for two distortion types - brightness and saturation across 5 distortion levels. In Fig. 8, we show results across five distortion levels for the first 12 distortions. The blue plot is the Feed-Forward ECE while the lower orange plot is its introspective counterpart. Apart from Level 5 contrast, intrsopective ResNet-18 is more calibrated than its feed-forward counterpart. This is in addition to the performance gains. The trend remains the same in the remaining distortions and among all considered networks. We average out ECE across 19 distortions and 5 challenge

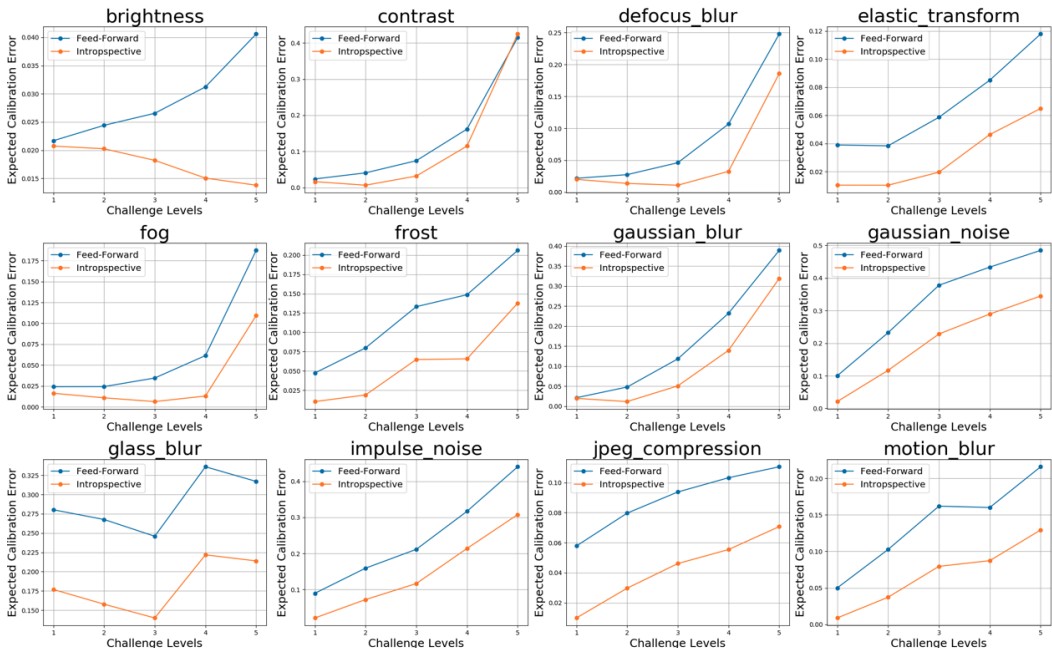

Figure 8: ECE vs distortion levels across 12 separate distortions from CIFAR-10C for ResNet-18.

levels and provide ECE results for ResNets-18, 34, 50, 101 in Table 5. Lower the error, better is the architecture. The proposed introspective framework decreases the ECE of its feed-forward backbone by approximately 42%. An additional metric called Maximum Calibration Error (MCE) is also used for comparison. While ECE averages out the calibration difference in all bins (From Section 6), MCE takes the maximum error among all bins Guo et al. (2017). The introspective networks outperform their feed-forward backbones among all architectures when compared using ECE and MCE.

Table 6: SimCLR and its supervised and introspective variations tested on CIFAR-10C.

| Methods | ResNet-18 | ResNet-34 | ResNet-50 | ResNet-101 |
|---|---|---|---|---|
| SimCLR Chen et al. (2020b) | 70.28% | 69.5% | 67.32% | 64.68% |
| SimCLR-MLP | 72.79% | 72.54% | 70.37% | 70.89% |
| SimCLR-Introspective (Proposed) | **73.32%** | **73.06%** | **71.28%** | **71.76%** |

## C.3    SIMCLR AND INTROSPECTION

SimCLR Chen et al. (2020b) is a self-supervised contrastive learning framework that is robust to noise distortions. The algorithm involves creating augmentations of existing data including blur, noise, rotations, and jitters. The network is made to contrast between all the augmentations of the image and other images in the batch. A separate network head $g(\cdot)$ is placed on top of the network to extract features and inference is made by creating a similarity matrix to a feature bank. Note that $g(\cdot)$ is a simple MLP. Our proposed framework is similar to SimCLR in that we extract features and use an MLP $\mathcal{H}(\cdot)$ to infer from these features. In Table 4, we show the results of Introspecting ResNets against SimCLR. However, this comparison is unfair since the features in SimCLR are trained in a self-supervised fashion. In this section, we train SimCLR for ResNets-18, 34, 50, 101 and train a new MLP $g(\cdot)$, not for extracting features, but to classify images. In other words, in Chen et al. (2020b), the authors create $g(\cdot)$ to be a $512 \times 128$ layer that extracts features. We train a network of the form $512 \times 128 - 128 \times 10$ that is trained to classify images. We then introspect on this $g(\cdot)$ to obtain $r_x$. Hence, our extracted features are a result of introspecting on self-supervision. Note that $g(\cdot)$ is now a fully supervised network. We pass CIFAR-10C through $g(\cdot)$ and name it SimCLR-MLP in Table 6. It is unsurprising that the fully-supervised SimCLR-MLP beats the self-supervised SimCLR across all four ResNets. The introspective network is called SimCLR-Introspective in Table 6. Note that there is less than 1% recognition performance increase across networks compared to SimCLR-MLP. Hence, the performance gains for introspecting on SimCLR-MLP is not as high as base ResNet

Table 7: Introspective Learning accuracies when $r_x$ is extracted with different loss functions for ResNet-18 on CIFAR-10C.

| Feed-Forward | MSE-M | CE | BCE | L1 | L1-M | Smooth L1 | Smooth L1-M | NLL | SoftMargin |
|---|---|---|---|---|---|---|---|---|---|
| 67.89% | **71.4%** | 69.47% | 70.76% | 70.12% | 70.72% | 70.42% | 70.63% | 70.93% | 70.91% |

architectures from Table 4. One hypothesis for this marginal increase is that the notions created within SimCLR-MLP are predominantly from the self-supervised features in SimCLR. These may not be amenable for the current framework of introspection that learns to contrast between classes and not between features within-classes.

### C.4 ABLATION STUDIES

The feature generation process in Section 2 is dependent on the loss function $J(\hat{y}, y)$. In this section, we analyze the performance of our framework for commonly used loss functions and show that the introspective network outperforms its feed-forward counterpart under any choice of $J(\hat{y}, y)$. We also ascertain the effect of the size of the parameter set in $\mathcal{H}(\cdot)$ on performance accuracy.

#### C.4.1 EFFECT OF LOSS FUNCTIONS

We extract $r_x$ using 9 loss functions and report the final distortion-wise level-wise averaged results Table 7. We do so for ResNet-18 and for the architecture of $\mathcal{H}(\cdot)$ shown in Table 3. The following loss functions are compared : CE is Cross Entropy, MSE is Mean Squared Error, L1 is Manhattan distance, Smooth L1 is the leaky extension of Manhattan distance, BCE is Binary Cross Entropy, and NLL is Negative Log Likelihood. Notice that the performance of $r_x$ extracted using all loss functions exceed that of the feed-forward performance. The shown results of MSE, L1-M and Smooth L1-M are obtained by backpropagating a $\mathbf{1}_N$ from Theorem 1 vector multiplied by the average of all maximum logits $M$, in the training dataset. We use $M$ instead of 1 because we want the network to be as confidant of the introspective label $y_I$ as it is with the prediction label $\hat{y}$. Note that the results in Table 7 are for CIFAR-10C. MSE-M outperforms NLL loss by $0.37\%$ in average accuracy and is used in our experiments.

#### C.4.2 EFFECT OF $\mathcal{H}(\cdot)$

We conduct ablation studies to empirically show the following : 1) the design of $\mathcal{H}(\cdot)$ does not significantly vary the introspective results, 2) the extra parameters in $\mathcal{H}(\cdot)$ are not the cause of increased performance accuracy.

**How does changing the structure of $\mathcal{H}(\cdot)$ change the performance?** We vary the architecture of $\mathcal{H}(\cdot)$ from a single linear layer to $4$ layers in the first half of Table 8 for ResNet-18. The results in the first three cases are similar. A four layered network performs worse than $f(\cdot)$. However, changing the weight decay from $5e^{-3}$ to $5e^{-4}$ during training increases the results to above $70\%$ but does not beat the smaller networks. For ResNet-18 architecture, the highest results are obtained when $\mathcal{H}(\cdot)$ is a 2-layered architecture but for the sake of uniformity, we use the results from a 3-layered network across all ResNet architectures.

**Are the extra parameters in $\mathcal{H}(\cdot)$ the only cause for increase in performance accuracy?** We show an ablations study of the effect of structure of $\mathcal{H}(\cdot)$ and $f(\cdot)$ on the introspective and feed-forward results in the second half of Table 3 on CIFAR-10-C dataset. The results are divided into four sections. In the first section, we show the performance of the original feed-forward network $f(\cdot)$, the performance when the final layer, $f_L(\cdot)$ is retrained using features $y_{feat}$ from Eq. 1, and the introspective network when $\mathcal{H}(\cdot)$ is a single layer. The second section shows the results when the features $y_{feat}$ are used to train a three layered network $f_L(\cdot)$, and the introspective network is also three layered. Finally, in section 3, we try to equate the number of parameters for $f_L(\cdot)$ and $\mathcal{H}(\cdot)$. Note that in all cases, $f_L(\cdot)$ and $\mathcal{H}(\cdot)$ are trained in the same manner as detailed in Section C.1. $\mathcal{H}(\cdot)$ beats the performance of $f_L(\cdot)$ among all ablation studies. Finally, similar to SimCLR, we forego using an MLP and use 10-Nearest Neighbors on $y_{feat}$ ($64 \times 1$) and $r_x$ ($640 \times 1$) for predictions. Both results are worse-off than their MLP results but $r_x$ outperforms $y_{feat}$.

Table 8: Ablation studies for $\mathcal{H}(\cdot)$ on CIFAR-10C.

| | **Part 1 : Varying the number of layers** | |
| --- | --- | --- |
| | Feed-Forward $64 \times 10$ | 67.89% |
| | $640 \times 10$ | 71% |
| R-18 | $640 \times 100 - 100 \times 10$ | **71.57%** |
| | $640 \times 300 - 300 \times 100 - 100 \times 10$ | 71.4% |
| | $640 \times 400 - 400 \times 200 - 200 \times 100 - 100 \times 10$ | 66.1% |
| | Feed-Forward $64 \times 10$ | 71.8% |
| R-50 | $2560 \times 300 - 300 \times 100 - 100 \times 10$ | **75.2%** |
| | $2560 \times 1000 - 1000 \times 500 - 500 \times 300 - 300 \times 100 - 100 \times 10$ | 73% |
| | **Part 2 : Is the performance increase only because of a large $\mathcal{H}(\cdot)$?** | |
| | Feed-Forward | 67.89% |
| | $f_L(\cdot)$ 1 Layer : $64 \times 10$ | 67.86% |
| | $\mathcal{H}(\cdot)$ 1 Layer : $640 \times 10$ | **71%** |
| | $f_L(\cdot)$ 3 Layers $64 \times 30 - 30 \times 20 - 20 \times 10$ | 63.61% |
| | $f_L(\cdot)$ 3 Layers $64 \times 512 - 512 \times 256 - 256 \times 10$ | 64.78% |
| R-18 | $\mathcal{H}(\cdot)$ 3 Layers: $640 \times 300 - 300 \times 100 - 100 \times 10$ | **71.4%** |
| | $f_L(\cdot)$, 6200 parameters : $64 \times 50 - 50 \times 40 - 40 \times 20 - 20 \times 10$ | 66.85% |
| | $\mathcal{H}(\cdot)$, 6400 parameters : $640 \times 10$ | **71%** |
| | Prediction on $y_{feat}$ using 10-NN (No $f_L(\cdot)$) | 66.31% |
| | Prediction on $r_x$ using 10-NN (No $\mathcal{H}(\cdot)$) | **68.76%** |
| | **Part 3 : VGG-16** | |
| | Feed-Forward | 68.96% |
| VGG-16 | $f(\cdot)$ $512 \times 1024 - 1024 \times 256 - 256 \times 10$ | 62.43% |
| | $\mathcal{H}(\cdot)$ $5120 \times 1000 - 1000 \times 100 - 100 \times 10$ | **73.79%** |

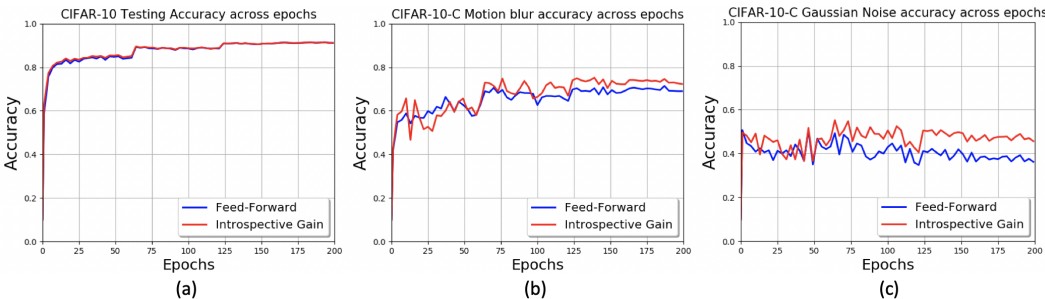

(a)   (b)   (c)

Figure 9: Introspective vs. Feed-Forward accuracy of ResNet-18 across training epochs on (a) CIFAR-10 original testset, (b) CIFAR-10C Motion Blur Testset on all 5 challenge levels, (c) CIFAR-10C Gaussian Noise Testset on all 5 challenge levels.

## C.5 INTROSPECTIVE ACCURACY ACROSS TRAINING EPOCHS

In Section 2, we make the assumption that $f(\cdot)$ is well trained to approximate $r_x$ using Theorem 1. In Section 3, the Fisher Vector analysis works when the gradients form distances across the manifold in $f(\cdot)$ which occurs if $f(\cdot)$ is well trained. In this section we show that, practically, introspection performs as well as feed-forward accuracy across training epochs on CIFAR-10 testset and outperforms feed-forward accuracy on CIFAR-10C distortions. We show results on original testset, gaussian noise and motion blur testsets in Fig. 9.

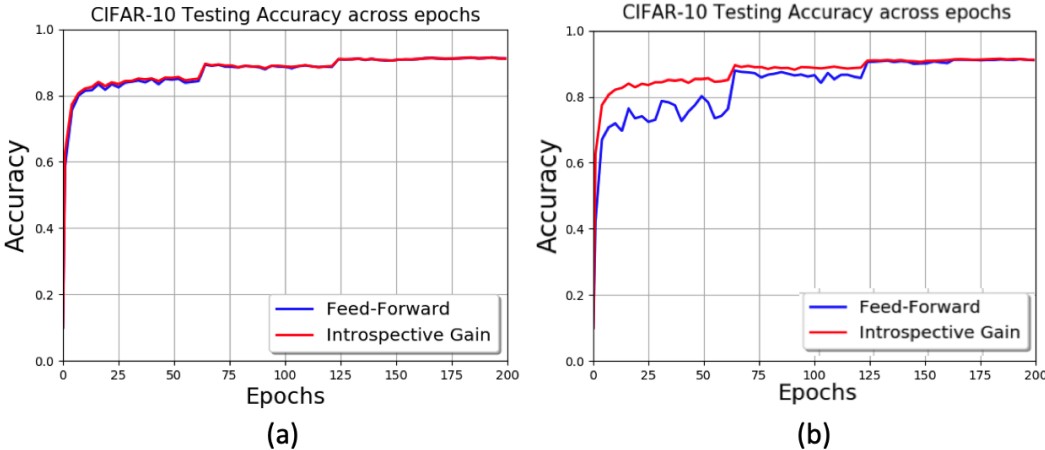

Figure 10: Introspective vs. Feed-Forward accuracy of ResNet-18 across training epochs when (a) $f(\cdot)$ and $\mathcal{H}(\cdot)$ are trained on the same training set (b) $\mathcal{H}(\cdot)$ is trained on a separate held-out validation set

**Training, Testing, and Results in Fig. 9a** In this experimental setup, ResNet-18 is trained for 200 epochs. The model states at multiples of 3 epochs from 1 to 200 are stored. This provides 67 states of $f(\cdot)$ along its training process. Each $f(\cdot)$ is tested on CIFAR-10 testset and the recognition accuracy is plotted in blue in Fig. 9a). The introspective features $r_x$ for all 67 states are extracted for the $50,000$ training samples. These $r_x$ are used to train 67 separate $\mathcal{H}(\cdot)$ of structure provided in Table 3 with a similar training setup as in Section C.1. The $r_x$ from the $10,000$ testing samples are extracted individually for each of the 67 $f(\cdot)$ states and tested. The results are plotted in red in Fig. 9a). Note the sharp spikes at epochs 60 and 120 where there is a change in the learning rate. Hence, when training and testing distributions are similar, introspective and feed-forward learning provides statistically similar performance across varying states of $f(\cdot)$.

**Training, Testing, and Results in Fig. 9b, c** We now consider the case when a network $f(\cdot)$ is trained on distribution $\mathcal{X}$ and tested on $\mathcal{X}'$ from CIFAR-10C distortions. The 67 trained models of ResNet-18 are tested on two distortions from CIFAR-10C. From the results in Fig. 3, introspective learning achieves one of its highest performance gains in Gaussian noise, and an average increase in motion blur after epoch 200. The results in Fig. 9 indicate that after approximately 60 epochs, the feed-forward network has sufficiently sensed notions to reflect between classes. This is seen in the performance gains in both the motion blur and Gaussian noise experiments.

**Training of $\mathcal{H}$ on a separate validation set in Fig. 10b** In all experiments, the introspective network $\mathcal{H}(\cdot)$ is trained on the same training set as $f(\cdot)$. In Fig. 10, we show the results when the introspective network is trained on a separate portion of the dataset. We use $40,000$ images to train $f(\cdot)$ and $10,000$ to train $\mathcal{H}(\cdot)$ both of which are randomly chosen. We follow the training procedure from before. The model states at multiples of 3 epochs from 1 to 200 are stored. This provides 67 states of $f(\cdot)$ along its training process. Each $f(\cdot)$ is tested on CIFAR-10 testset and the recognition accuracy is plotted in blue in Fig. 10b). The $\mathcal{H}(\cdot)$ at each iteration on the other hand is trained with the $10,000$ images. However, it has access to the notions created from the remaining $40,000$ images and hence the results for introspection match Fig. 9a) which is reproduced in Fig. 10a). The feed-forward results catch up to the introspective results around epoch 60. At Epoch 120, we add back the $10,000$ held-out images into the training set of $f(\cdot)$ and the results match between Fig. 10a) and Fig. 10b).

## C.6 DOMAIN ADAPTATION ON OFFICE DATASET

In Section 3, we claim that introspection helps a network to better classify distributions that it has not seen while training. In Section 5, we tested on 95 new distributions in CIFAR-10C and 30 new distributions in CIFAR-10-CURE. In this section, we evaluate the efficacy of introspection when there is a domian shift between training and testing data under changes in background, and camera

Table 9: Performance of Proposed Introspective $\mathcal{H}(\cdot)$ vs Feed-Forward $f(\cdot)$ Learning under Domain Shift on Office dataset

| Architectures | | DSLR ↓ Amazon | DSLR ↓ Webcam | Amazon ↓ DSLR | Amazon ↓ Webcam | Webcam ↓ DSLR | Webcam ↓ Amazon |
|---|---|---|---|---|---|---|---|
| ResNet-18 | $f(\cdot)$ | 39.1 | 78 | 62.9 | 59 | 89.8 | 42.2 |
| (%) | $\mathcal{H}(\cdot)$ | **47** | **90.7** | **67.3** | **63.9** | **96** | **44** |
| ResNet-34 | $f(\cdot)$ | 41.8 | 83.3 | **67.3** | 60.1 | 90.6 | 41.7 |
| (%) | $\mathcal{H}(\cdot)$ | **46.4** | **89.8** | **67.3** | **63.9** | **97.8** | **43.3** |
| ResNet-50 | $f(\cdot)$ | - | - | 67.3 | 62 | 92.4 | 33.4 |
| (%) | $\mathcal{H}(\cdot)$ | - | - | **78.1** | **68.4** | **97.8** | 30.8 |
| ResNet-101 | $f(\cdot)$ | - | - | 62.9 | 59 | 89.8 | 31.77 |
| (%) | $\mathcal{H}(\cdot)$ | - | - | **76.5** | **67.3** | **92.4** | **33.6** |

Table 10: Performance of Proposed Introspective $\mathcal{H}(\cdot)$ vs Feed-Forward $f(\cdot)$ Learning under Domain Shift on VisDA Dataset

| ResNet-18 | Plane | Cycle | Bus | Car | Horse | Knife | Bike | Person | Plant | Skate | Train | Truck | All |
|---|---|---|---|---|---|---|---|---|---|---|---|---|---|
| $f(\cdot)$ (%) | 27.6 | 7.2 | **38.1** | 54.8 | 43.3 | **4.2** | **72.7** | **8.3** | 28.7 | 22.5 | **87.2** | 2.9 | 38.1 |
| $\mathcal{H}(\cdot)$ (%) | **39.9** | **27.6** | 19.6 | **79.9** | **73.5** | 2.7 | 46.6 | 6.5 | **43.8** | **30** | 73.6 | **4.3** | **43.58** |

acquisition setup among others. Specifically, the robust recognition performance of $\mathcal{H}(\cdot)$ is validated on Office Saenko et al. (2010) dataset using Top-1 accuracy. The Office dataset has 3 domains - images taken from either Webcam or DSLR, and extracted from Amazon website. Images can belong to any of 31 classes and they are of varying sizes - upto $1920 \times 1080$. Hence, results on Office shows the applicability of introspection on large resolution images. ImageNet pre-trained ResNet-18,34,50,101 He et al. (2016) architectures are used for $f(\cdot)$. The final layer is retrained using the source domain while the remaining two domains are for testing. The experimental setup, the same detailed in Section 5, is applied and the Top-1 accuracy is calculated. The results are summarized in Table 9. In every instance, the top domain is $\mathcal{X}$ - the training distribution, and the bottom domain is $\mathcal{X}'$ - the testing distribution. Note that ResNet-50 and 101 failed to train on 498 images in DSLR source domain. The results of introspection exceed that of feed-forward learning in all but ResNet-50 when classifying between Webcam and Amazon domains.

### C.7 Domain Adaptation on Vis-DA dataset

Validation results on a synthetic-to-real domain shift dataset called VisDA Peng et al. (2017) are presented in Table 10. VisDA has 12 classes with about $152,000$ synthetic training images, and $55,000$ real validation images. The validation images are cropped images from MS-COCO. ResNet-18 architecture pretrained on ImageNet is finetuned on the synthetically generated training images from VisDA dataset fro 200 epochs. It is then tested on the validation images and the recognition performance is shown in Table 10 as feed-forward $f(\cdot)$ results. Introspective $\mathcal{H}(\cdot)$ results are obtained and shown when $f(\cdot)$ is ResNet-18. There is an overall improvement of $5.48\%$ in terms of performance accuracy. However, the individual class accuracies leave room for improvement.

### C.8 Active Learning

In Table 1, the mean recognition accuracies across the first 20 rounds of Active Learning experiments for commonly used query strategies are shown. We plot these recognition accuracies across for all five query strategies in Fig. 11. The x-axis is the round at which the performance is calculated. The calculated accuracy is plotted on the y-axis. The experiment starts with a random 100 images in round 1. Each strategy queries using either a round-wise sample trained $f(\cdot)$ or a round-wise sample trained $\mathcal{H}(\cdot)$. Note that at each round, the networks are retrained. This continues for 20 rounds. Both BALD Gal et al. (2017) and BADGE Ash et al. (2019) applied on $\mathcal{H}(\cdot)$ consistently beat its $f(\cdot)$ counterpart on every round. This is because both these methods rely on extracting features from the network as compared to the other three techniques that directly use the output logits from either $\mathcal{H}(\cdot)$ or $f(\cdot)$. Since the network is not well-trained at the initial stages - due to a dearth of training data - the introspective network is not as consistent as the feed-forward network among Entropy, Least

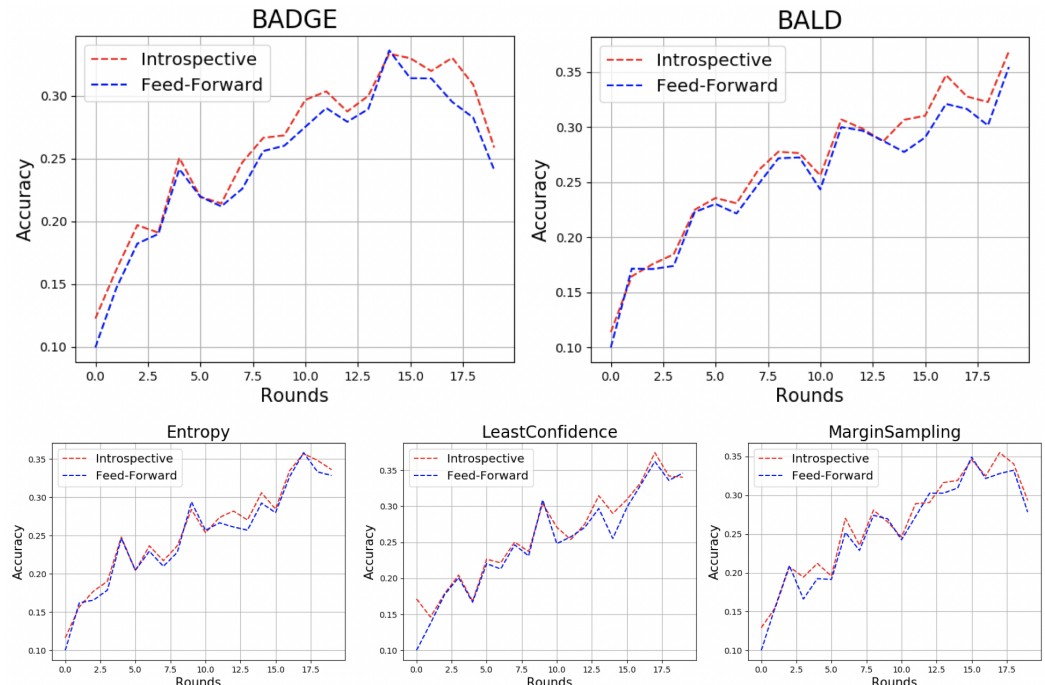

Figure 11: Introspective vs. Feed-Forward accuracy of ResNet-18 across training rounds for state-of-the-art techniques in an active learning setting. The query batch size per round is 1000. The trainset is CIFAR-10 and testset is Gaussian Noise from CIFAR-10C.

Table 11: Out-of-distribution Detection of existing techniques compared between feed-forward and introspective networks.

| Methods | OOD Datasets | FPR (95% at TPR) ↓ | Detection Error ↓ | AUROC ↑ |
|---|---|---|---|---|
| | | Feed-Forward/Introspective | | |
| MSP Hendrycks & Gimpel (2016) | Textures | 58.74/**19.66** | 18.04/**7.49** | 88.56/**97.79** |
| | SVHN | 61.41/**51.27** | 16.92/**15.67** | 89.39/**91.2** |
| | Places365 | 58.04/**54.43** | 17.01/**15.07** | 89.39/**91.3** |
| | LSUN-C | **27.95**/27.5 | **9.42**/10.29 | **96.07**/95.73 |
| ODIN Liang et al. (2017) | Textures | 52.3/**9.31** | 22.17/**6.12** | 84.91/**91.9** |
| | SVHN | 66.81/**48.52** | 23.51/**15.86** | 83.52/**91.07** |
| | Places-365 | **42.21**/51.87 | 16.23/**15.71** | **91.06**/90.95 |
| | LSUN-C | **6.59**/23.66 | **5.54**/10.2 | **98.74**/ 95.87 |

Confidence, and Margin strategies. Nonetheless, $\mathcal{H}(\cdot)$ outperforms $f(\cdot)$ on average across all rounds.

## C.9 OOD

**Adversarial setting in Table 2** A datapoint $z$, is perturbed as $z + \epsilon$ and the goal of the detector is to classify $z \in \mathcal{X}$ or $z \in \mathcal{X}'$. This modality is proposed by the authors in Chen et al. (2020a) and we use their setup. PGD attack with perturbation $0.0014$ is used. The same MSP and ODIN detectors from Table 2 are utilized. On 4 OOD datasets, both MSP and ODIN show a performance gain across all three metrics on $\mathcal{H}(\cdot)$ compared to $f(\cdot)$. Note that the results in Table 11 is for ResNet-18 architecture for the same $f(\cdot)$ and $\mathcal{H}(\cdot)$ used in other experiments including Fig. 2.

**Vanilla setting in Table 11** In Table 11, we show the results of out-of-distribution detection when $\mathcal{X}$ is CIFAR-10 and $\mathcal{X}'$ are the four considered datasets. Note that among the four datasets, textures and SVHN are more out-of-distribution from CIFAR-10 than the natural image datasets of Places365 and LSUN. The results of the introspective network is highest on Textures DTD dataset.

Table 12: Performance of Contrastive Features against Feed-Forward Features and other Image Quality Estimators. Top 2 results in each row are highlighted.

| Database | PSNR HA | IW SSIM | SR SIM | FSIMc | Per SIM | CSV | SUM MER | Feed-Forward UNIQUE | Introspective UNIQUE |
|---|---|---|---|---|---|---|---|---|---|
| **Outlier Ratio (OR, ↓)** | | | | | | | | | |
| **MULTI** | 0.013 | 0.013 | **0.000** | 0.016 | 0.004 | **0.000** | **0.000** | **0.000** | **0.000** |
| **TID13** | **0.615** | 0.701 | 0.632 | 0.728 | 0.655 | 0.687 | **0.620** | 0.640 | **0.620** |
| **Root Mean Square Error (RMSE, ↓)** | | | | | | | | | |
| **MULTI** | 11.320 | 10.049 | 8.686 | 10.794 | 9.898 | 9.895 | **8.212** | 9.258 | **7.943** |
| **TID13** | 0.652 | 0.688 | 0.619 | 0.687 | 0.643 | 0.647 | 0.630 | **0.615** | **0.596** |
| **Pearson Linear Correlation Coefficient (PLCC, ↑)** | | | | | | | | | |
| **MULTI** | 0.801 | 0.847 | 0.888 | 0.821 | 0.852 | 0.852 | **0.901** | 0.872 | **0.908** |
| | -1 | -1 | 0 | -1 | -1 | -1 | -1 | -1 | |
| **TID13** | 0.851 | 0.832 | 0.866 | 0.832 | 0.855 | 0.853 | 0.861 | **0.869** | **0.877** |
| | -1 | -1 | 0 | -1 | -1 | -1 | 0 | | 0 |
| **Spearman's Rank Correlation Coefficient (SRCC, ↑)** | | | | | | | | | |
| **MULTI** | 0.715 | **0.884** | 0.867 | 0.867 | 0.818 | 0.849 | **0.884** | 0.867 | **0.887** |
| | -1 | 0 | 0 | 0 | -1 | -1 | 0 | 0 | |
| **TID13** | 0.847 | 0.778 | 0.807 | 0.851 | 0.854 | 0.846 | 0.856 | **0.860** | **0.865** |
| | -1 | -1 | -1 | -1 | 0 | -1 | 0 | 0 | |
| **Kendall's Rank Correlation Coefficient (KRCC)** | | | | | | | | | |
| **MULTI** | 0.532 | **0.702** | 0.678 | 0.677 | 0.624 | 0.655 | 0.698 | 0.679 | **0.702** |
| | -1 | 0 | 0 | 0 | -1 | 0 | 0 | 0 | |
| **TID13** | 0.666 | 0.598 | 0.641 | 0.667 | **0.678** | 0.654 | 0.667 | 0.667 | **0.677** |
| | 0 | -1 | -1 | 0 | 0 | 0 | 0 | 0 | |

## C.10   IMAGE QUALITY ASSESSMENT

**Related Works**   Multiple methods have been proposed to predict the subjective quality of images including PSNR-HA (Ponomarenko et al., 2011), IW-SSIM (Wang & Li, 2011), SR-SIM (Zhang & Li, 2012), FSIMc (Zhang et al., 2011), PERSIM (Temel & AlRegib, 2015), CSV (Temel & AlRegib, 2016), SUMMER (Temel & AlRegib, 2019), and UNIQUE (Temel et al., 2016). All these methods extract structure related hand-crafted features from both reference and distorted images and compare them to predict the quality. Recently, machine learning models have been proposed to directly extract features from images (Temel et al., 2016). Temel et al. (2016) propose UNIQUE that uses a sparse autoencoder trained on ImageNet to extract features from both reference and distorted images. We use UNIQUE as our base network $f(\cdot)$.

**Feed-Forward UNIQUE**   Temel et al. (2016) train a sparse autoencoder with a one layer encoder and decoder and a sigmoid non-linearity on $100,000$ patches of size $8 \times 8 \times 3$ extracted from ImageNet testset. The autoencoder is trained with MSE reconstruction loss. This network is our $f(\cdot)$. UNIQUE follows a full reference IQA workflow which assumes access to both reference and distorted images while estimating quality. The reference and distorted images are converted to YGCr color space and converted to $8 \times 8 \times 3$ patches. These patches are mean subtracted and ZCA whitened before being passed through the trained encoder. The activations of all reference patches in the latent space are extracted and concatenated. Activations lesser than a threshold of $0.025$ are suppressed to $0$. The choice of threshold $0.025$ is made based on the sparsity coefficient used during training. Similar procedure is followed for distorted image patches. The suppressed and concatenated features of both the reference and distorted images are compared using Spearman correlation. The resultant is the feed-forward estimated quality of the distorted image.

**Introspective-UNIQUE**   We use the architecture and the workflow from Temel et al. (2016) which is based on feed-forward learning to demonstrate the value of introspection. We replace the feed-forward features with the proposed introspective features. The loss in Eq. 6 for introspection is not between classes but between the image $x$ and its reconstruction $\tilde{x}$ from the sparse autoencoder from Temel et al. (2016). For a reference image $x$, $r_x$ is derived using $J(x, \tilde{x})$. Hence, gradients of $r_x$ span the space of reconstruction noise. Since the need in IQA is to characterize distortions, we obtain $r_x$ for reference images from the first layer and project both reference and distorted images onto $r_x$. These projections are compared using Spearman correlation to assign a quality estimate. In this setting, $\mathcal{H}(\cdot)$ is the projection operator and Spearman correlation. Hence, Introspective-UNIQUE broadens introspection in the following ways - 1) defining introspection on generative models, 2) using gradients in the earlier layers of a network.

**Results**   We report the results of the proposed introspective model in comparison with commonly cited methods Table 12. We utilize MULTI-LIVE (MULTI) (Jayaraman et al., 2012) and TID2013 (Ponomarenko et al., 2015) datasets for evaluation. The performance is validated using outlier ratio (consistency), root mean square error (accuracy), Pearson correlation (linearity), Spearman correlation (rank), and Kendall correlation (rank). Arrows next to each metric in Table 12 indicate the desirability of a higher number ($\uparrow$) or a lower number($\downarrow$). Two best performing methods for each metric are highlighted. The proposed framework is always in the top two methods for both datasets in all evaluation metrics. In particular, it achieves the best performance for all the categories except in OR and KRCC in TID2013 dataset. The feed-forward model does not achieve the best performance for any of the metrics in MULTI dataset. However, the same network using introspective features significantly improves the performance and achieves the best performance on all metrics. For instance, the feed-forward model is the third best performing method in MULTI dataset in terms of RMSE, PLCC, SRCC, and KRCC. However, the introspective features improve the performance for those metrics by 1.315, 0.036, 0.020, and 0.023, respectively and achieve the best performance for all metrics. This further reinforces the plug-in capability of the proposed introspective inference.

