# OpenReview forum: "Introspective Learning : A Two-Stage approach for Inference in Neural Networks"
_ICLR.cc/2022/Conference — ICLR 2022 Submitted_

### Official Review · Reviewer_8FMs · 2021-11-01

**Correctness:** 2
**Technical Novelty And Significance:** 2
**Empirical Novelty And Significance:** Not applicable
**Recommendation:** 3
**Confidence:** 4

**Main Review:**

The reading of the paper does not flow properly. The methods are hard to understand and are not technically sound. Many core elements of the math discourse are introduced abruptly, such as $L$ and $\mathcal{H}$. The definition of introspection is vague and not formal. The gap between the intuition about introspection presented in the introduction and the practical realization described in the methods sections remains wide and leaves the readers without a proper foundation to build their understanding. For instance, the intuition of introspection the authors give in the introduction and in Figure 1 calls for a reflection stage applied only to some promising alternative classes, while the proposed method applies the reflection stage to every single class.
Moreover, I think that some less relevant sections should be shortened and others more relevant should be deepened. For instance, given the nature of the scientific contribution, this paper does not require in my opinion such an extensive Application section, while spending more words on making clear the methods in the middle sections and the general idea in the Introduction could provide more value to the reader.

Yet, to me, the idea of enriching neural networks features with gradient information through a two-stage inference process seems novel and sounds interesting. Results suggest that this research direction can be relevant, so I encourage the authors to pursue this work and focus on communicating it properly.

In the following, I list some comments about specific parts of the paper:

INTRODUCTION
- I believe that philosophical references can be of great value to the ML and AI community. In this sense I appreciate the reference to Locke, I wish the authors have expanded more this point.
- "For the application of recognition... ...EQUATION (1)" This part is not clear. I do not understand why the authors use the projection concept in this context. $L$ seems a tag to indicate the final layer but then becomes a natural number ($L-1$). The first equation about $y_{feat}$ has a reference to $y$ that does not appear in the equation.

INTROSPECTIVE FEATURES
- The authors use the term explanations without defining what is an explanation for them and justify that definition. It is a textual string? A heat map on an image? Why we can consider heat maps as *explanations*?
- The definition of introspection is very vague. "...is the measurement of change induced in the network parameters...", explicit this measurement, state the quantities. The reader here is expecting a formal definition but this definition is not formal.

INTROSPECTIVE FEATURE VISUALIZATION
- "...Grad-CAM indicates that the pink and round body, and straight beak are the reasons for the decision. [...] the network highlights the neck of the spoonbill to indicate that since S-shaped neck is not observed, x cannot be a flamingo. ..." I believe that here authors are overinterpreting the heat-maps. How can we say that the heat-map is caused by shapes rather than patterns? The clean high-level concepts of neck, beak, straight, S-shaped are what the network sees or what the authors see?

INTROSPECTIVE FEATURE EXTRACTION
- The assumption that a forward and backward pass through f is O(1) does not sound familiar to me. There is a dependence with respect to the size of the input.
- Probably this subsection could be shorter, and at least part of the computational cost consideration could go in the appendix.

INTROSPECTIVE NETWORK
- H appears suddenly and everything is built on it. This is one of the more obscure parts of the paper.
- The assumption of a mean squared error loss is not trivial, we usually do not train classifiers with the mean squared error loss but with cross-entropy.
- "having networks that introspect consecutively, by creating a trade-off" is not clear.
-  let the reader grasp how the reflection stage can fix the first inference of the sensing stage, spend some words on this case or maybe provide an example.

EXPERIMENTS
- "Given a data distribution $x \in X$" why distribution?
- "10 separate bins based on their prediction confidence". Considering the outputs of a neural network as confidences is problematic, this point should be expanded. DNNs are known to have unreliable uncertainty estimates (MacKay, 1995, Szegedy et al., 2014, Nguyen et al., 2015)

**Summary Of The Paper:**

The authors present a new inference pipeline for neural networks $f$ trained to solve classification problems. They augment the features processed by $f$ with the loss gradients with respect to the weights of the last layer. Their pipeline requires two steps: the first one coincides with a standard evaluation of $f(x)$, while gradients are evaluated and processed in the second step to derive the final prediction. Experiments show slight improvements in classification accuracy and cuts of calibration errors.

**Summary Of The Review:**

The core parts of the paper are not clear and do not allow a proper comprehension and evaluation of the scientific contribution of this work.
The overall idea seems valuable and experimental results appear promising. Yet, with the current manuscript, I can only judge the potential of this work rather than the work itself, so at this stage my recommendation is a 3: reject, not good enough.

---

> ### Author Response · Authors · 2021-11-25
> **Revisions to improve paper comprehension**
>
> We thank the reviewer for their comments. We have made substantial revisions to the paper, based on the reviewer's recommendations. The major revisions are in blue. Here, we address the reviewer comments (in bold).
>
> **The reading of the paper does not flow properly. The methods are hard to understand and are not technically sound. Many core elements of the math discourse are introduced abruptly, such as $\mathcal{L}$ and $\mathcal{H}(\cdot)$. The definition of introspection is vague and not formal. The gap between the intuition about introspection presented in the introduction and the practical realization described in the methods sections remains wide and leaves the readers without a proper foundation to build their understanding.**
>
> The majority of the revisions made are to address these comments:
>
> 1. We have redone Fig.1 to include both a high level motivation and a method-based framework. The reflection stage, feature intuitions, and $\mathcal{H}(\cdot)$ are all presented in the figure. We have removed Fig.2 due to its redundancy and have placed it in the appendix along with its generation. This allows for a substantial discussion about $\mathcal{H}(\cdot)$ in the Introduction itself.
>
> 2. We believe that introducing $\mathcal{H}(\cdot)$ and $r_x$ in the Introduction bridges the gap between the high level motivation and the methodology. We also make changes to Section 2, specifically Section 2.1. We use Lemma 1 to motivate the conditioning of $\mathcal{H}(\cdot)$ on all introspective features. Hence, the noise has to break the relationship between $N$ pairwise logits to alter the prediction.
>
> 3.In addition to presenting Lemma 1 and Theorem 1 as equations, we present visualizations in Appendix B.1, Fig. 5. These provide credence to the acceleration aspects of the method.
>
> **The intuition of introspection the authors give in the introduction and in Figure 1 calls for a reflection stage applied only to some promising alternative classes, while the proposed method applies the reflection stage to every single class.**
>
> In practice, when the data is noisy, obtaining the *promising alternative classes* is problematic since the original prediction may itself be incorrect. Instead, by conditioning $\mathcal{H}(\cdot)$ on all class differences, we overcome this problem. We make this explicit in the Introduction. We state the following,
>
> 1. *Note that there can be $N$ such features for a sensing network $f(\cdot)$ trained to differentiate between $N$ classes. These features are then combined to obtain the final introspective feature $r_x$.*
>
> 2. *Not only should the network sense the feed-forward patterns, it must also satisfy $\mathcal{H}(\cdot)$'s $N$ notions of differences. In this paper, we show that the inference process is more generalizable due to these $N$ additional inferential constraints.*
>
>  **Moreover, I think that some less relevant sections should be shortened and others more relevant should be deepened. For instance, given the nature of the scientific contribution, this paper does not require in my opinion such an extensive Application section, while spending more words on making clear the methods in the middle sections and the general idea in the Introduction could provide more value to the reader.**
>
> We thank the reviewer for these constructive comments. Reviewers Q4tj and DxqE also had similar comments regarding the method and acceleration. We have considerably shortened the acceleration, increased the intuition in the Introduction and connected the high level Section 1 with Section 3 through a new Section 2.1. To do so we moved Fig.2 and its explanations and generations to Appendix A.
>
> **I believe that philosophical references can be of great value to the ML and AI community. In this sense I appreciate the reference to Locke, I wish the authors have expanded more this point.**
>
> We are glad that the reviewer appreciates this and we have added a new Appendix A that deals with the philosophical points of introspection. We inspect introspective features as answers to reasoning-based questions, specifically Abductive reasoning. Not only does this provide a high level motivation, but it also depicts some possible future directions with respect to reasoning-based workflows that are not just end-to-end.
>
> Reviewer DxqE also brings up the point for need of introspection in Recognition. We provide a biological plausibility for the need for introspection and connect it to our results as well.
>
> **"For the application of recognition... ...EQUATION (1)" This part is not clear. I do not understand why the authors use the projection concept in this context. $L$  seems a tag to indicate the final layer but then becomes a natural number $(L-1)$. The first equation about  has a reference to $y$ that does not appear in the equation.**
>
> We have revised the uploaded manuscript to make this clearer. We have removed references to projection and explicitly define $f_L$ and $f_{L-1}$ as the last and penultimate layers.

---

> > ### Author Response · Authors · 2021-11-25
> > **Continuation of the previous comment: Revisions to improve paper comprehension**
> >
> > **The authors use the term explanations without defining what is an explanation for them and justify that definition. It is a textual string? A heat map on an image? Why we can consider heat maps as explanations?**
> >
> > We use the term explanations as visual *post-hoc* explanations - visual heat maps that justify decisions. This is made explicit as a footnote in Page 2.
> >
> > **"...Grad-CAM indicates that the pink and round body, and straight beak are the reasons for the decision. [...] the network highlights the neck of the spoonbill to indicate that since S-shaped neck is not observed, x cannot be a flamingo. ..." I believe that here authors are overinterpreting the heat-maps. How can we say that the heat-map is caused by shapes rather than patterns? The clean high-level concepts of neck, beak, straight, S-shaped are what the network sees or what the authors see?**
> >
> > We understand the reviewer's concerns about *post-hoc* interpretability. We have softened these claim and explicitly state in the footnote on Page 2 that these explanations are *post-hoc* and require human interpretation. However, we believe that from a high level motivation in our new Fig.1, these heat maps provide the required intuition for what the $r_i$ features represent - a network's notion of the class differences. We only use these in Introduction and not in Section 2 as we were doing before.
> >
> > **The definition of introspection is very vague. "...is the measurement of change induced in the network parameters...", explicit this measurement, state the quantities. The reader here is expecting a formal definition but this definition is not formal.**
> >
> > We have done so in the revised uploaded version.
> >
> > **The assumption that a forward and backward pass through f is O(1) does not sound familiar to me. There is a dependence with respect to the size of the input.**
> >
> > We use gradients from the final fully connected layer only. Hence, in practice, we store the activations from the penultimate layer. The activations pass through the final layer to provide $\hat{y}$. The loss passes backward through the same layer for us to obtain $r_x$, hence making both passes $\mathcal{O}(1)$. This is made explicit in the notation in Page 3 after Eq.5.
> >
> > **Probably this subsection could be shorter, and at least part of the computational cost consideration could go in the appendix.**
> >
> > We did so and also differentiated Lemma 1. which is used to motivate the robustness of features, (Section 2.1) with Theorem 1, which is used for acceleration.
> >
> > **H appears suddenly and everything is built on it. This is one of the more obscure parts of the paper.**
> >
> > We have fixed this with the revision to the Introduction and Fig.1.
> >
> > **The assumption of a mean squared error loss is not trivial, we usually do not train classifiers with the mean squared error loss but with cross-entropy.**
> >
> > We take the initial analysis from the bias-variance tradeoff and the calibration analysis presented in [Kuleshov & Liang (2015) in paper]. It has been our observation that in the literature, the common technique has been to analyze for MSE even when applying other losses. We also use cross entropy loss for training both $f(\cdot)$ and $\mathcal{H}(\cdot)$. The training details are in the Appendix and we will explicit this in the manuscript itself.
> >
> > **"having networks that introspect consecutively, by creating a trade-off" is not clear.**
> >
> > We have rewritten this statement to make it more explicit - we mean that further introspection by extracting gradients from $\mathcal{H}(\cdot)$ does not seem to provide additional gains.
> >
> > **let the reader grasp how the reflection stage can fix the first inference of the sensing stage, spend some words on this case or maybe provide an example.**
> >
> > Section 2.1 has been rewritten to provide this intuition. Moreover, The Generalization of $\mathcal{X'}$ using $\mathcal{H}(\cdot)$ in Section 3 is a Fisher Vector interpretation of why the introspective features generalize the predictions from sensing.
> >
> > **"Given a data distribution " why distribution?**
> >
> > This should be : Given a data sample $x \in \mathcal{X}$. We missed making this correction in the revision and will do so in the final version.
> >
> > **"10 separate bins based on their prediction confidence". Considering the outputs of a neural network as confidences is problematic, this point should be expanded. DNNs are known to have unreliable uncertainty estimates (MacKay, 1995, Szegedy et al., 2014, Nguyen et al., 2015)**
> >
> > We thank the reviewer for this comment. We have added a new section to the Related works - Confidence and Uncertainty. We include these references. More importantly we use this section to motivate introspection for both the Active Learning and OOD Detection applications.
> >
> > We apologize for not engaging with the reviewer sooner. We hope that the revisions and response is in time and the reviewer can decide based on it. We will be monitoring this space for additional comments and can make any required changes.

---

### Official Review · Reviewer_Q4tj · 2021-11-02

**Correctness:** 2
**Technical Novelty And Significance:** 3
**Empirical Novelty And Significance:** 3
**Recommendation:** 5
**Confidence:** 4

**Main Review:**

The paper proposes an interesting and novel approach for improving robustness and calibration under distribution shift.  Results on CIFAR10-Corrupted are very promising, and the authors further show the method is able to improve in active learning and OOD detection experiments when inputs are corrupted with noise.

Lemma 1: assumptions should be stated precisely in the lemma statement itself. As it stands, the lemma statement makes it seem like the statement holds exactly, which seems very unreasonable at first glance.
Regarding the reasonableness of the assumptions, a  "well trained" network might have high confidence predictions on input data it was trained on, but not necessarily on test data. Empirically, it has been observed that test data (particularly OOD test data) tends to have much higher entropy in predictions. Regarding theorem 1, can you test how well this approximation holds? In CIFAR10, there are only 10 classes, so it's reasonable to actually compute and store the full gradient features (especially since only the last layer is used). It would be good to see how a classifier performs using the full set of introspective features instead of the gradient wrt to the all ones vector. I would also highly recommend experiments to see if introspection (and this approximation in particular) scale well to more complex tasks with more classes like CIFAR100 or Imagenet, as I believe the current set of experiments to be quite limited.

**Questions and Suggested Ablations:**

Can you elaborate on why we should expect the introspective features to provide benefits when the new classifier is trained on the exact same training data as the base network? It seems very plausible to me that the gradients on training points the base
network has already overfitted heavily are a very different distribution than the gradients on previously unseen points, so I wouldn't necessarily have expected the introspective classifier to generalize well. I'm also curious what would happen if the introspection network is trained on a separate validation set.

How does this compare to directly taking the activations of the network instead of gradients? The final activations should be very closely related to the gradients of the last layer, differing by multiplicative terms corresponding to derivatives of the loss wrt to the output logits. This experiment would help to examine whether the structure of the gradient information from each class over possibly some benefits of just retraining model components on top of a pretrained model features.

Did you examine introspective features from other layers?

In the OOD detection, are the same adversarial images used for the feedforward and introspective network, or is the adversarial image for the introspective method specifically targeted at the extra network using introspective features?

Some missing related work:

Gradients as features: [2] is a crucial piece of missing related work, which also considers using gradients of a network as features, and for example show some similar results in recovering the original classifier performance on the trianing set. It would also be good to discuss and reference [1], which similarly uses gradients of base models as features, but applies it towards finetuning for transfer learning.

Conditional/predictive normalized maximum likelihood: Conditional/predictive normalized maxmimum likelihood [3, 4] considers retraining models for each label of the test input and combining these models' predictions to output a final prediction, essentially introspecting for each possible label to get predictions. While this work differs by using introspection to provide features for a separate classifier, it would be good to mention these other works as applying similar ideas of utilizing information on how model's change for specific labels of the test input in order improve predictions in the face of distribution shift.

**Misc comments:**

There's a bit of a weird jump in the assumptions in the paper: lemma 1 assumes cross-entropy loss, but in sec 3, we're assuming network is trained with MSE for the derivation.

In addition to showing ECE for CIFAR10-C, I would also recommend including uncertainty-aware metrics like NLL and Brier score in the appendix, as these are proper scoring rules.


**Citations:**

[1] Zinkevich, Martin A. et al. “Holographic Feature Representations of Deep Networks.” UAI (2017).

[2] Mu, Fangzhou, Yingyu Liang, and Yin Li. "Gradients as features for deep representation learning." ICLR (2020)

[3] Zhou, Aurick, and Sergey Levine. "Amortized Conditional Normalized Maximum Likelihood: Reliable Out of Distribution Uncertainty Estimation." International Conference on Machine Learning (2021)

[4] Bibas, Koby, Yaniv Fogel, and Meir Feder. "Deep pnml: Predictive normalized maximum likelihood for deep neural networks." arXiv preprint arXiv:1904.12286 (2019).


**Summary Of The Paper:**

The paper proposes introspective networks, a technique utilizing the gradients of a trained base model to make more robust predictions when faced with distribution shift.

**Summary Of The Review:**

Overall, the paper presents an interesting approach with promising results for improving robustness, both in terms of accuracy and uncertainty estimation. I believe with a more clear presentation, more thorough discussion of related work, more extensive experimental evaluations, and ablations/evaluations of specific assumptions and design choices, this will be a valuable contribution.

---

> ### Author Response · Authors · 2021-11-26
> **Ablations and Related Works**
>
> We thank the reviewer for their comments. We have made substantial revisions to the paper, based on the reviewer's recommendations. The major revisions are in blue. Here, we address the reviewer comments (in bold).
>
> Related Works
> -----------------
>
> We thank the reviewer for the pertinent references. We include all these references in the Related Works. We substantially expand the Related Works section with two new paragraphs and increase the Two-stage Architectures paragraph with the new references. These are used to motivate the ablation studies in Table 8 and the applications in Section 6. Changes in the revised manuscript are in blue.
>
> Ablations Studies
> ---------------------
> **Regarding the reasonableness of the assumptions, a "well trained" network might have high confidence predictions on input data it was trained on, but not necessarily on test data. Empirically, it has been observed that test data (particularly OOD test data) tends to have much higher entropy in predictions.**
>
> We use two ways to test our approximation: 1) Visualizing gradients directly, 2) Empirical results
>
> 1) In Fig.5, Appendix B.1, we simulate the *well-trained* aspect of a neural network on MNIST dataset. On MNIST, a network performs with an accuracy of 99% on test data. We train a neural network with a final fully connected layer of $50\times 10$. We then backpropagate for each possible introspective label and plot them. We show six such plots in Fig.5. Each plot is a $50\times 10$ matrix of all gradients. For an input image 5, we see in the visualizations that the only values that are present in the gradients are the prediction class $\hat{y}$ and the introspective class $y_I$.  This validates Eq.4 from Lemma 1 in the revised version - the loss is only dependent on the logits $y_{\hat{y}}$ and $y_I$. Hence, we only take the gradients at the $y_I^{th}$ location.
>
> 2) On MNIST dataset with Blur distortions, the feed-forward accuracy decreases to $60%$ accuracy. With our introspective framework, we can increase the results by more than $12\%$. We do not present these results in the paper because of the simple characteristics of the MNIST dataset.
>
> The visualizations on CIFAR-10 are not as clean as MNIST. However, the theory still stands and is validated in the empirical results in Sections 4 and 5.
>
> **Regarding theorem 1, can you test how well this approximation holds? In CIFAR10, there are only 10 classes, so it's reasonable to actually compute and store the full gradient features (especially since only the last layer is used). It would be good to see how a classifier performs using the full set of introspective features instead of the gradient wrt to the all ones vector.**
>
> In our experiments on CIFAR-10, each introspective matrix without any approximation is $640\times 10$. For $10$ classes, each $r_x$ is a vector of size $64,000\times 1$. We were not able train $\mathcal{H}(\cdot)$ using these features. However, by reducing each introspective matrix from $640\times 10$ to $640\times 1$ using Lemma $1$, we obtain $r_x$ as a concatenation of $10$ vectors to obtain $r_x$ of size $6400\times 1$. Training $\mathcal{H}(\cdot)$ using these features provided statistically insignificant differences to our results after Theorem 1. Even with a higher entropy, we believe the difference between two classes is sufficient to encapsulate introspection, atleast to account for noise on CIFAR-10C.
>
> **I would also highly recommend experiments to see if introspection (and this approximation in particular) scale well to more complex tasks with more classes like CIFAR100 or Imagenet, as I believe the current set of experiments to be quite limited.**
>
> One of the challenges in the current implementation is the size of the $r_x$ and its dependence on the number of classes $N$. We acknowledge this under Limitations in Section 7. In Appendix C.9, Table 9, we provide results on Office dataset. This dataset has 32 classes and large resolution images. Our IQA experiments in Table 12 are conducted on HD resolution images. In both cases, introspection provides accuracy gains.
>
> **Can you elaborate on why we should expect the introspective features to provide benefits when the new classifier is trained on the exact same training data as the base network?**
>
> We revised the manuscript and provide this intuition in Section 2.1. On a feed-forward network, noise has to change the relationship between the maximum logit and the nearest logit to change the network's prediction. In our framework, we obtain $N$ separate relationships between class differences and condition $\mathcal{H}(\cdot)$ on all these differences. Since the relationships are pairwise orthogonal, the noise has to break these $N$ relationships thereby making the introspective framework more robust to the same noise. Lemma 1 does not change this intuition - however, it makes it easier for the noise to break the relationships because of less dimensions. Empirically, this does not change the results.

---

> > ### Author Response · Authors · 2021-11-26
> > **Continuation of the previous comment : Ablation Studies**
> >
> > **I'm also curious what would happen if the introspection network is trained on a separate validation set.**
> >
> > We thank the reviewer for this ablation study. We perform this ablation study in Fig 10, Appendix C.5. The base network $f(\cdot)$ is trained on 40,000 images from the training set and $\mathcal{H}(\cdot)$ on only 10,000 images. Since the introspective network has access to the original training data through $f(\cdot)$, it performs statistically similarly as if it were trained on all $50,000$ images. We show results across 200 epochs. However, results for $f(\cdot)$ suffers. After epoch 120, we add the 10,000 images back into the training set of $f(\cdot)$.
> >
> > Since the goal of this paper is to validate the potential of introspection on feed-forward networks, we believe $f(\cdot)$ is disadvantaged by not having access to all data. However, for future works this can be very interesting in understanding $\mathcal{H}(\cdot)$'s performance under limited data for transfer learning purposes.
> >
> > **How does this compare to directly taking the activations of the network instead of gradients? The final activations should be very closely related to the gradients of the last layer, differing by multiplicative terms corresponding to derivatives of the loss wrt to the output logits. This experiment would help to examine whether the structure of the gradient information from each class over possibly some benefits of just retraining model components on top of a pretrained model features.**
> >
> > We thank the reviewer for these comments. We perform extensive ablation studies in Table 8 including - training a 1-layer new network on activations, training 3-layer network on activations to match the introspective framework, comparing networks of same parameter sizes between 4-layer feed-forward and 3-layer introspective networks, and foregoing MLP to use Nearest Neighbor classifier on activations and gradients. In all settings, the extra class-pairwise information provided by the gradients is useful to create robust decisions.
> >
> > We also use the reference provided by the reviewer [1], to address this issue. We add this within the Related works section.
> >
> > **Did you examine introspective features from other layers?**
> >
> > We did. For the purpose of recognition, the results were worse than the feed-forward network. The results from the penultimate fully-connected layer and the last convolution layer were extracted and used to train separate networks. In both cases, the gradient features did not have the requisite class dependency information to provide good results. Moreover, Lemma 1 and Theorem 1 are not applicable in these scenarios. Hence, the features were very large which impeded the training of the introspective network.
> >
> > However, for the application of IQA, where the goal is to introspect directly on the image features rather than the classes, we use gradients of an Autoencoder architecture directly after the bottleneck layer. We describe this and specifically state it in Appendix C.10, last line in Page 26.
> >
> > **In the OOD detection, are the same adversarial images used for the feedforward and introspective network, or is the adversarial image for the introspective method specifically targeted at the extra network using introspective features?**
> >
> > The original network $f(\cdot)$ is trained on pristine data. So is the introspective network $\mathcal{H}(\cdot)$. In the testing stage, adversarial images are created for the base network $f(\cdot)$ and used to provide the feed-forward results. The same adversarial images are used to obtain the $r_x$ features by introspecting using the wrong prediction. Hence, while the adversarial images are not created for $\mathcal{H}(\cdot)$, they alter the underlying prediction for introspection. This setting is taken from [Chen et al. (2020a) in paper].
> >
> > We also have the vanilla OOD framework without adversarial noise in Table 11. If the reviewer feels that this setting is fairer, we can bring this table into the main paper and move the adversarial table to the appendix. The conclusions will still be the same - more OOD a dataset is, better is the capability of introspection to detect it.
> >
> > **There's a bit of a weird jump in the assumptions in the paper: lemma 1 assumes cross-entropy loss, but in sec 3, we're assuming network is trained with MSE for the derivation.**
> >
> > Consistent with literature when it comes to Bias-Variance Tradeoff, we use MSE for analysis. However, we train all networks using cross entropy. This is stated in Appendix and we will make it clearer in the manuscript.
> >
> > **In addition to showing ECE for CIFAR10-C, I would also recommend including uncertainty-aware metrics like NLL and Brier score in the appendix, as these are proper scoring rules.**
> >
> > We added Brier score to the appendix but the results are not better than feed-forward. We are currently looking into this.
> >
> > [1] Mu, Fangzhou, Yingyu Liang, and Yin Li. "Gradients as features for deep representation learning." ICLR (2020)

---

### Official Review · Reviewer_DxqE · 2021-11-03

**Correctness:** 3
**Technical Novelty And Significance:** 3
**Empirical Novelty And Significance:** 2
**Recommendation:** 6
**Confidence:** 3

**Main Review:**

# Strengths
* An interesting modification to standard neural networks for object recognition with improvements in generalization and  calibration
* Extensive experiments showing the usefulness of the method in multiple applications

# Weaknesses
* Confusing explanation of the proposed method lacking a visual representation of the full model architecture, its components, and what the introspective features represent
* Model is evaluated only on CIFAR-10 related datasets which are very low resolution images and few possible classes
* According to the authors, the proposed method does not scale well for larger datasets. This makes its applicability very limited as CIFAR-10 is only a toy dataset
* Some controls seem rather arbitrary and a better choice of parameters for the alternative baselines could have been done

# Detailed review
## Role of introspection for object recognition
Object recognition in humans is an extremely fast visual computation, taking place within ~200ms of stimulus presentation, and it is unlikely to depend on introspection. It is not clear the motivation of why including introspective learning in artificial neural networks would improve their performance. Furthermore, standard neural networks are typically optimized to classify an object as belonging to one class and not belonging to the others (all classes are used in the cost function).  In that sense, it could be argued that they also are trained to answer the questions illustrated in Figure 1.

## Confusing explanation of the proposed methodology
Unfortunately, the authors could have done a better job at explaining their method. In my opinion, some of the mathematical details could be moved to the appendix, and in their place, the authors should focus on clarifying the training and inference process of the introspective networks. A visual representation of the full model architecture, its components, what the introspective features represent, and how the model is trained would be extremely helpful to the reader. Because of this it was not clear where the computational complexity of the introspective networks come from. The MLP used is very standard and the introspective features are obtained from the model’s last layer gradients.

## Poor choice of controls
Since adding the 3 layers to the standard model made it considerably worse, the choice of using 3 additional layers on the control for the same number of parameters was a poor choice by the authors. The MLP in the introspective network has a higher number of features but only a single layer. A better comparison would be to make the last layer of the standard model with more features, keeping only 1 layer, or keeping the same penultimate layer and adding a new much wider layer to compensate the smaller number of parameters.

## Minor comments
* Fonts in several figures are very small. Please improve their readability
* In Figure 4b and Figure 6, the x-axis should be corruption severity and go from 1 to 5
* Figure 2 does not add much and could go to supplementary
* Figure 1 is very high-level and in its place, the authors should provide a more concrete visualization of their method


**Summary Of The Paper:**

In this paper, the authors propose a modification to standard neural networks used for object classification tasks to incorporate what they call “introspective learning”. This consists on training a multi-layer perceptron (MLP) on the neural network introspective features. These are obtained by calculating the gradients over the last layer weights of the model on a loss function corresponding to posing an introspective question: why is the correct label A instead of B? The authors apply this procedure to several neural networks of the ResNet model family and show that the introspective-networks have better generalization to distributional shifts and smaller calibration errors on datasets with the same classes and image sizes as CIFAR-10. Finally, they show further improvements in multiple applications.

**Summary Of The Review:**

The authors propose a very interesting modification to neural networks for object recognition and do a convincing job in showing that it improves generalization and calibration. However, the proposed method is only tested in CIFAR-10 and its use in larger networks and datasets is questionable. Furthermore, the explanation of the method is unnecessarily confusing and better controls could have been chosen to make a more convincing case.

---

> ### Author Response · Authors · 2021-11-25
> **Introspection process, object recognition, and control experiments**
>
> We thank the reviewer for the feedback. We apologize for not engaging with the reviewer sooner. We have made changes to the manuscript and uploaded the revised version. The major changes are in blue. We specifically address some of the comments by the reviewer below. The reviewer comments are boldfaced.
>
> **Unfortunately, the authors could have done a better job at explaining their method. In my opinion, some of the mathematical details could be moved to the appendix, and in their place, the authors should focus on clarifying the training and inference process of the introspective networks.**
>
> **Figure 2 does not add much and could go to supplementary**
>
> **Figure 1 is very high-level and in its place, the authors should provide a more concrete visualization of their method**
>
> 1. *Regarding Figs.1 and 2* : We thank the reviewer for this feedback. We have moved the explanations in Fig.2 to Appendix A. We have redone Fig.1 to include both a high level motivation of our method (the *Not Detect* features of introspection, and the goal of the reflection stage) as well as the methodology (the introspective network's inputs and outputs, and the sensing network itself).
>
> 2. *Explanation of the method* : By bringing in $\mathcal{H}(\cdot)$ within Fig.1, we introduce and motivate the introspective network in the Introduction itself. We also motivate this in Section 2. We start Section 3 by re-referencing Fig.1. The exact details of training the network are provided in Section 5 and Appendix C.1.
>
> 3. *Mathematical details* : The computational aspect of the method is now moved to the appendix. In its place, we use the math to motivate our method in Section 2.1. We use Lemma 1 to motivate the conditioning of $\mathcal{H}(\cdot)$ on all introspective features. Hence, the noise has to break the relationship between $N$ pairwise logits to change the performance.
>
> 4. The computational complexity is not in training the MLP network itself but rather in extracting the features. We make this clearer in the manuscript.
>
> **Object recognition in humans is an extremely fast visual computation, taking place within ~200ms of stimulus presentation, and it is unlikely to depend on introspection. It is not clear the motivation of why including introspective learning in artificial neural networks would improve their performance.**
>
> We thank the reviewer for this question. The reviewer is correct in this statement - recognition is fast and mostly a feed-forward process. However, when there is uncertainty involved - either due to distributional shift or noise - we tend to reason about our decisions. We add a reasoning-based motivation in Appendix A. We interpret introspective features as hypotheses that answer contrastive questions. Moreover, our results also support this - when train and testsets are from the same distribution, there is no change in results (Section 5, Paragraph 4). However, when there is a distributional difference, we notice the gains for introspection - on CIFAR-10C, CIFAR-10-CURE, active learning, OOD and IQA experiments. Moreover, higher the distributional difference, larger is the introspective gain - Fig.6b.
>
> From a biological perspective, introspection derives inspiration from works on saliency and particularly expectancy mismatch [1]. As the reviewer points out, human visual system detects salient portions of an image and attends to them in a feed-forward process. An alternative perspective to this is expectancy-mismatch - the idea that HVS attends to those features that deviate from expectations. We simulate this via introspection. By asking *Why P, rather than Q?*, we ask the network to examine its expectations and describe the mismatches. This is seen as the lack of S-shaped neck in spoonbill in Fig.1. Recently, there are other works that explore this, for neural networks [2].
>
> We will include this as a motivating factor for introspection in Appendix A. We realize that this process works better during the discussion phase but due to a few unavoidable circumstances, we could not engage before. We hope that the reviewer considers these changes.
>
> **A better comparison would be to make the last layer of the standard model with more features, keeping only 1 layer, or keeping the same penultimate layer and adding a new much wider layer to compensate the smaller number of parameters.**
>
> We add a wider layer after the penultimate in Table 8. The results we found were similar as before. We performed the same experiment on VGG-16 which has a bigger penultimate layer - it is 512x10 compared to a ResNet-18's 64x10. We present this as well in Table 8 and the results and conclusions are similar.
>
> [1] Summerfield, Christopher, and Tobias Egner. "Expectation (and attention) in visual cognition." Trends in cognitive sciences 13.9 (2009): 403-409.
>
> [2] Sun, Yutong, Mohit Prabhushankar, and Ghassan AlRegib. "Implicit Saliency in Deep Neural Networks." 2020 IEEE International Conference on Image Processing (ICIP). IEEE, 2020.

---

> > ### Comment · Reviewer_DxqE · 2021-11-30
> > **Slight improvements to the paper but no updated score**
> >
> > I thank the authors for the time to address some of my concerns and the changes to the paper. While these changes improve the quality of the paper, I maintain my original assessment of the paper.

---

### Official Review · Reviewer_yYCU · 2021-11-08

**Correctness:** 3
**Technical Novelty And Significance:** 3
**Empirical Novelty And Significance:** 3
**Recommendation:** 6
**Confidence:** 3

**Main Review:**

Vanilla feedforward networks are single-stage, filtering the inputs as they pass forward through the network and predicting the class of the input in the final layer. Motivated by the human tendency to not only gather sensory data, but to reflect on those data in the face of uncertainty to arrive at a classification, the authors introduce a second, introspective or reflective, stage.

This stage is defined as the change in the parameters in the network if a different label were assigned to the network's input, and can be measured as a gradient with respect to the network weights of a loss function.  After showing how the calculation can be simplified in space and time, they then demonstrate how introspection can improve the generalizability of classifications to distributions that are shifted from the original training sets and can improve the calibration of the network, the confidence of the classification minus the accuracy.  They finally demonstrate how the approach can be used in a number of different applications.

Overall, the introspective approach is, to the reviewer's knowledge, novel and potentially of interest as a general method for improving the performance of feed-forward networks. The authors provide the reader with intuition (e.g., Fig 2), rigor, and empirical data (Fig 3/4).  One potential issue that arises in this empirical data is that introspection does not improve the accuracy of ResNet-18 on CIFAR-10C, as they state, although the ECE decreases systematically.

Unfortunately, it is difficult to judge by eye whether the differences in Fig 4 are statistically significant.  The authors should include error bars or some other visual cue that indicate the uncertainty of the plotted values.  Additionally, without a statistically significant result, it is difficult to convince the reader that the authors' approach will work in a meaningful manner empirically in cases of interest.

If possible, I would suggest the authors find such an example, or, if they cannot, to perhaps indicate future work that might improve the technique in such a way that it could in fact do so. Beyond this, the paper is written and organized well and clearly.

**Summary Of The Paper:**

The authors present a novel 2-stage technique to improve the classification accuracy of feedforward ANNs.  In particular, after the feedforward pass, a so-called introspective stage occurs, the goal of which is to ascertain why the particular class label was provided rather than a different label.  This stage, which tends to improve the accuracy of the class predictions, is modular and can be added onto networks under varying task conditions.

**Summary Of The Review:**

The authors present a new method, introspection, that can be added on top of feedforward networks to improve accuracy and generalization of the networks.  While they have written a well-structured paper that provides both intuition and rigor, the empirical results are somewhat lacking in that there are no statistically significant results that support the claims of accuracy improvements that introspection can obtain.

---

> ### Author Response · Authors · 2021-11-25
> **Additional results and visualizations**
>
> We thank the reviewer for their feedback. We apologize for not engaging with the reviewer sooner. We have made changes to the manuscript and we request that the reviewer consider these changes in the final decision. We respond directly to the comments raised by the reviewer here.
>
> **Unfortunately, it is difficult to judge by eye whether the differences in Fig 4 are statistically significant. The authors should include error bars or some other visual cue that indicate the uncertainty of the plotted values. Additionally, without a statistically significant result, it is difficult to convince the reader that the authors' approach will work in a meaningful manner empirically in cases of interest.**
>
> 1. Our goal in Fig.4 is to demonstrate that across all 19 distortions, introspection provides an increase in results. This increase, is however, varying based on the distortion characteristics. We analyze this variance based on distortions themselves - distortions that affect global characteristics like in brightness and contrast do not propagate to the last layer and hence are not conducive for introspection.
>
> 2. In this regard, we believe, Fig.6 in the appendix C.2 is a better representation of the results. On CIFAR-10C, ResNet-18 provides 4.2% increase in accuracy. A ResNet-18 with introspection matches the results from a base feed-forward ResNet-50. We highlight this in C.2.1. The difference in results across distortion levels is also provided in Fig.6b. As the distortion increases, the contribution of introspection increases as well.
>
> 3. In Fig.7, we provide results specifically for ResNet-18 across distortions and across levels. Here the increase in results on Level 4 and 5 distortions is apparent. The average increase across all distortions in Level 5 is above 8%. On 190,000 images in Level 5 test set, this relates to about 15,200 additional correct recognitions compared to feed-forward model. On the entire test dataset, of 950,000 images, introspection classifies an additional 33,000 images correctly compared to the base ResNet-18 model making it statistically significant.
>
> 4. We also show the results of introspection as a generalizability tool. This is done by using it as a plug-in approach on existing robustness techniques in Table 4 and Table 6. In Table 9, we show results on Office dataset where we do not use any test domain data for training and yet achieve a performance accuracy boost.
>
> We believe that the new visualizations provide a better benchmark of our results. Moreover, we provide a statistical significance test on Image Quality Assessment experiment in IQA, since the datasets are quite small - 3000 images in TID and 225 images in MULTI-LIVE. This is provided in Table 12. Our results are statistically significant in 24 of the 48 compared metrics against other techniques. In no metric is it statistically insignificant.

---

### Author Response · Authors · 2021-11-23
**Summary of Revision**

We thank the Reviewers for their feedback and comments. Due to unavoidable circumstances, we were unable to engage during the discussion phase. However, we have addressed each of the raised points. The revised paper is uploaded and the the changes are in blue. A summary of the major revisions are as follows:

1. We have altered Fig.1 to explain both the high level motivation as well as the methodology as per the reviewer comments.
2. We have introduced $\mathcal{H}(\cdot)$ in the Introduction itself and motivate it within both Sections 1 and 2.
3. To make room for additional discussions regarding features and network, we have moved Fig.2 and its discussion to the appendix, as per reviewer comments.
4. Lemma 1, instead of being only an acceleration technique, is also used to motivate generalizability in Section 2.1.
5. We have expanded the Related Works section to include Gradients as Features, and Uncertainty and Confidence sections. We motivate the downstream tasks within these sections.

In the Appendix, the following major revisions have been made:

1. We add a philosophical component to introspection by considering the features from a reasoning perspective.
2. We add Fig.5 for MNIST dataset. This comes closest to simulating a well-trained network. The gradients from the final layer are visualized which shows the efficacy of Lemma 1.

 Again, we are sorry for the late response and hope the Reviewers have a chance to look through the revised version.

---

### Decision · Program_Chairs · 2022-01-20

**Decision:**

Reject

**Comment:**

The reviewers all appreciated the novel concept behind the work. I agree with this, I think the principles behind the work are novel and interesting, and I would encourage the authors to improve the validation of this method and publish it in the future.

However, reviewers also raised a number of issues with the current paper: (1) the evaluation appears a bit preliminary, and could be improved significantly with additional datasets and more ablations/comparisons; (2) it's not clear if the improvements from the method are especially significant; (3) the writing could be improved (I do see that the authors made a significant number of changes and improved parts of the paper in response to reviewer concerns to a degree). Probably the writing issues could be fixed, but the skepticism about the experiment results seems harder to address, and while I recognize that the authors made an effort to point some existing ablations in the paper that do address parts of what the reviewers raised, I do think that in the balance the experimental results leave the validation of the work as somewhat borderline.

While less important for the decision, I found that the paper is somewhat overselling the contribution in the opening -- while the particular concept of using gradients as features in this way is interesting, similar ideas have been proposed in the past, and the paper would probably be better if it was more clearly positioned in the context of prior work rather than trying to present a new "framework" like this. It kind of feels like it's biting off too much in the opening, and then delivering a comparatively more modest (but novel and interesting!) technical component.